# Predicting gene regulatory networks from cell atlases

Andreas Fønss Møller[1] , Kedar Nath Natarajan[1,2]

**Recent single-cell RNA-sequencing atlases have surveyed and identified major cell types across different mouse tissues. Here, we computationally reconstruct gene regulatory networks from three major mouse cell atlases to capture functional regulators critical for cell identity, while accounting for a variety of technical differences, including sampled tissues, sequencing depth, and author assigned cell type labels. Extracting the regulatory crosstalk from mouse atlases, we identify and distinguish global regulons active in multiple cell types from specialised cell type–specific regulons. We demonstrate that regulon activities accurately distinguish individual cell types, despite differences between individual atlases. We generate an integrated network that further uncovers regulon modules with coordinated activities critical for cell types, and validate modules using available experimental data. Inferring regulatory networks during myeloid differentiation from wild-type and Irf8 KO cells, we uncover functional contribution of Irf8 regulon activity and composition towards monocyte lineage. Our analysis provides an avenue to further extract and integrate the regulatory crosstalk from single-cell expression data.**

## Introduction

Multicellular organisms are composed of different tissues consisting of varied cell types that are regulated at the single-cell level. Single-cell RNA sequencing (scRNA-seq) enables high-throughput gene expression measurements for unbiased and comprehensive classification of cell types and factors that contribute to individual cell states (1, 2). The underlying expression heterogeneity between single cells can be attributed to finer grouping of cell types, inherent stochasticity and variations in underlying functional and regulatory crosstalk (3, 4, 5, 6). Single cells maintain their cell state and also respond to a variety of external cues by modulating transcriptional changes, which are governed by complex gene-regulatory networks (GRNs) (7, 8). A GRN is a specific combination of transcription factors (TFs) and co-factors that interact with cis-regulatory genomic regions to mediate a specialised transcriptional programme within individual cells (9, 10). Briefly, a regulon is a collection of a TF and all its transcriptional target genes. The GRNs define and govern individual cell type definition, transcriptional states, spatial patterning and responses to signalling, and cell fate cues (11). Recent computational approaches have enabled inference of the gene regulatory circuitry from scRNA-seq datasets (9, 12, 13, 14, 15, 16).

Recently two major single-cell mouse atlases studies were published (17, 18). The Tabula Muris (TM) and Mouse Cell Atlas (MCA), profiled >500,000 individual single cells using three different scRNA-seq platforms, across multiple murine tissues to provide a broad survey of constituent cell types and gene expression patterns and thereby demarcating shared and unique signatures across single cells. The three cell atlases use different scRNA-seq platforms and technologies including Smart-seq2 (TM-SS2: (19)), 10× Chromium (TM-10×: (20)), and Microwell-seq (18).

For regulatory and mechanistic insights beyond cell type survey across the three atlases, we have to extend analysis beyond comparison of gene expression patterns. The computational inference of TFs and their regulated gene sets (regulons) provides an avenue to extract the regulatory crosstalk from single-cell expression data (9, 10, 21, 22). Here, we set out to comprehensively reconstruct GRNs from single-cell atlases and address the following questions: (i) Which TFs, master regulators, and co-factors (i.e., regulons) govern tissue and cell types? (ii) Do inferred regulons regulate "specific" or multiple cell types? (iii) Which regulons and regulated gene sets are critical for individual cell identity?

In our integrative analysis, we identify regulon modules that globally regulate multiple cell groups and tissues across cell atlases. The cell type–specific regulons are characterised by distinct composition and activity, critical for their definition. We find that regulons and their activity scores are robust indicators of cell type identity across cell atlases, irrespective of composition differences. We uncover modules of regulons and reconstruct an integrated atlas-scale regulatory network, and also validate network interactions using available experimental datasets. Importantly, we uncover the functional consequence of Irf8 regulon perturbation at the single-cell level during myeloid lineage decisions from wild-type and Irf8 knockout cells. We uncover a distinctly depleted Irf8 regulon composition and activity of Irf8 knockouts, validating the specification bias from monocytes to granulocytes. This work provides a consensus view

---

[1]Department of Biochemistry and Molecular Biology, Functional Genomics and Metabolism Unit, University of Southern Denmark, Odense, Denmark [2]Danish Institute of Advanced Study, University of Southern Denmark, Odense, Denmark

Correspondence: knn@bmb.sdu.dk

of key regulators functioning in different cell types that define cellular programs at the single-cell level.

## Results

To identify regulatory networks across the different mouse cell types and tissues, we analysed both "TM" and "MCA" scRNA-seq studies (17, 18). The TM contains >130,000 annotated single cells profiled using two scRNA-seq methods (referred as atlases), full-length Smart-seq2 (~54k single cells, 18 tissues, and 81 cell types), and 3′-end droplet based 10× Chromium (70,000 single cells, 12 tissues, and 55 cell types). The MCA contains >230,000 annotated single cells profiled using the author's 3′-end microwell-seq platform (38 tissues, 760 cell types Supplemental Data 1).

We aimed to integrate the atlases to identify cell type–specific regulons and build a consensus regulon atlas (Fig 1A; detailed workflow in Fig S1). As each atlas samples different mouse tissues and scRNA-seq technologies (full length versus 3′ end) to identify hundreds of varied cell types across cellular resolutions (discussed below), a fundamental challenge is to effectively link the original author's cell type annotation across cell atlases. We address the challenge of integrating cell type classification by combining two complementary approaches. First, we manually devised a generalised vocabulary consisting of broadly defined "7 cell groups" for and standardise annotation between cell atlases (three datasets). Second, we utilize scMAP, an unsupervised scRNA-seq cell projection method (23), to link the original author's cell type annotation across cell atlases Supplemental Data 1. By using TM-10× Chromium annotations as a reference and by combining both approaches, our generalised vocabulary contains "7 cell groups" consisting of "55 reference cell types." The seven cell groups include *Immune* (22 subgroups), *Specialised* (12 subgroups), *Epithelial* (7 subgroups), *Stem* (4 subgroups), *Endothelial* (4 subgroups), *Basal* (3 subgroups), and *Blood* (3 subgroups) (Fig S2A). Subsequently, we applied our two-step approach to individual atlases, that is, TM-10× (Fig S2B), TM Smart-seq2 (TM-SS2; Fig S3A), MCA (Fig S3B), and to all atlases integrated together (Fig S4). Our approach allows us to build and link an integrated mouse atlas consisting of 831-author assigned unique cell type labels from 50 tissues to a consensus of 55 reference cell types and 7 cell groups (Fig S4, the Materials and Methods section, and Table S1).

We support the robustness of our generalised vocabulary and projection mapping approach by multiple analysis. Across individual tissues, we re-confirmed that author cell type labels are robustly mapped to reference cell types and cell groups both in individual and integrated atlas (Fig S5A liver, Fig S5B spleen, Figs S2A and S3A and B). The individual atlases have technical difference owing to the different number of cells profiled (Fig S6A top panel), sequencing depth (library size, Fig S6A middle panel), number of tissues profiled (12 TM-10×, 18 TM-SS2, 38 MCA; Figs S2B and S3A and B), scRNA-seq chemistry (Full-length versus 3′), scRNA-seq platform, and number of genes detected (Fig S6A bottom panel). The dropout distribution for individual atlases highlights the relationship between the number of cells profiled, library size, and genes detected (Fig S6B). Specifically, MCA compared with TM atlases has the highest number of profiled cells at sparse sequencing depth, lower gene detected, and highest dropout rates across reference cell groups (Fig S6A and B). Our seven reference cell groups have high and proportional number of cells from both integrated (Fig S6C) and individual atlas (Fig S6D). For example, the immune cell group consists of 20,133 individual cells classified across 22 reference cell types, whereas the blood cell group consists of 1,559 cells classified into three reference cell types (Figs S6C and S2A). We further present the different technical features for each reference cell type across integrated and individual atlas (Fig S7). Our two-step approach consisting of simplified cell group and subgroup classification allows us to mitigate technical and cell type label discrepancies and integrate mouse cell atlases to investigate global and specific regulators across atlases.

Feature selection is a crucial aspect for robust regulon inference and composition. We tested a variety of different feature sets for both integrated and individual cell atlases. We selected a reasonable cutoff of genes detected in at least 10% of all single cells, consisting of 11,245 overlapping genes across three atlases (Fig 1B). This cutoff robustly and proportionally captures the reference cell groups across integrated and individual atlases, despite the technical differences (Fig S7). To infer GRNs, we applied SCENIC, a framework for network inference, reconstruction, and clustering from scRNA-seq data (10). The SCENIC framework is applied directly on the single-cell expression matrix combining (i) "GRNBoost" for identification of TFs and co-expressed genes from single-cell expression matrix, (ii) "RcisTarget" for defining "regulons" (i.e., enriched and validated TFs with their direct downstream target genes containing annotated motif, and prunes co-expressed indirect targets), and (iii) "AUCell" for scoring regulon activity ([RAS] regulon activity scores) in single cells. Our motivation for using SCENIC for atlas scale regulon inference was threefold. Firstly, SCENIC identifies, scores direct TF-target interactions, while pruning indirect and co-expressed connections. The RCisTarget cross-matches regulons with known TF-target databases, as opposed to de-novo predictions, and infers, scores both TF–TF and TF-target. Second, SCENIC does not prerequisite a single-cell trajectory/pseudotime (12) and is suited to atlas-scale analysis. Third, SCENIC tools (GENIE3/GRNBoost) are ranked highly in a recent benchmarking study (12). Applying SCENIC, we identify 279 unique regulons, with >60% (174 regulons) shared across the three atlases (Fig 1B). The high degree of regulon overlap between the three atlases, in spite of technical differences, highlights that single-cell regulatory state is predominantly governed by core set regulators and their activities within individual cells. A recent study also applied SCENIC, but only for MCA data using only the author-assigned cell type labels (21).

To distinguish the regulatory activity within individual cells, we performed dimensionality reduction using UMAP on RASs of ~250,000 single cells, integrating all atlases (24). We coloured individual cells using the reference cell groups (Fig 1C), predicted cell cycle stage (Fig 1C, right), and tissue of origin (Fig S8A). We observe good visual separation between the seven cell groups based on RAS, highlighting robustness of cell group classification and ability of RAS to distinguish functional cell types in integrated atlas. The overlapping cell groups are biologically and functionally related, with similar RAS and tissue origin. For example, a subset of immune

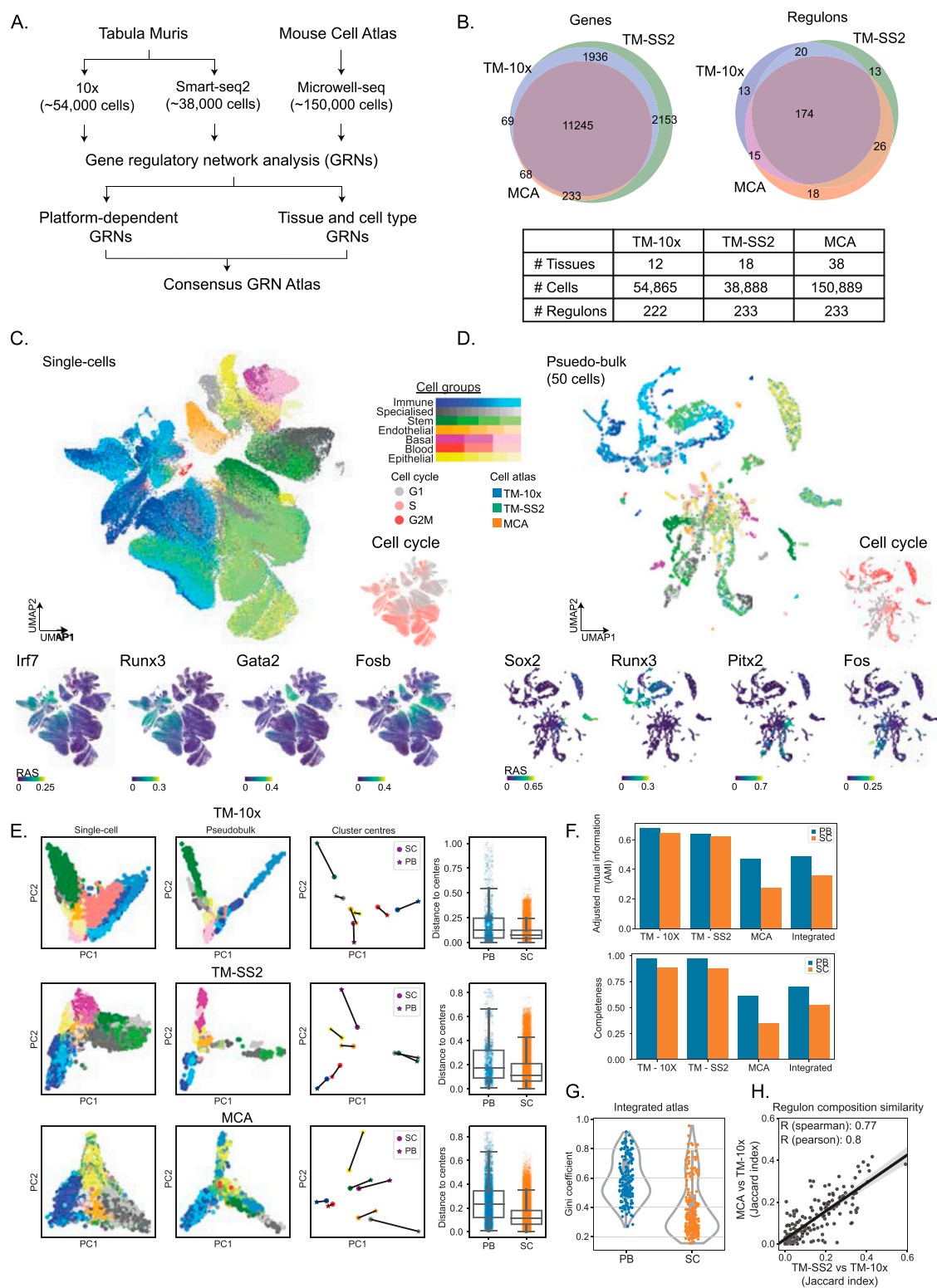

**Figure 1. Gene regulatory inference from integrated single-cell transcriptomic atlases. (A)** Overview of datasets and analysis performed in this study. **(B)** Venn plots and table representation of shared and unique features across cell atlases including tissues, number of cells, and regulons across cells. We used 11,245 overlapping genes and resulting 279 unique regulons for regulatory analysis. **(C)** UMAP embedding of single cells (centre) based on regulon activity scores (RAS) from integrated mouse atlases. The individual cells are coloured by 55 reference cell types corresponding to seven cell groups. The surrounding plots highlight examples of individual regulons (Irf7, Runx3, Gata2, and Fosb) coloured by RAS, predicted cell cycle stages (right), and overlaid on UMAP. **(D)** UMAP embedding of 50-cell pseudobulk samples, based on RAS from integrated mouse atlases. The surrounding plots highlight examples of individual regulons (Sox2, Runx3, Pitx2, and Fos) coloured by RAS, predicted cell

(blue) and stem (green) cell groups originating from bone marrow overlap in the integrated atlas (bottom left: Figs 1C and S8A). The cell cycle stage prediction based on scRNA-seq is also consistent with cell groups and reference cell type classification (25, 26). As expected, most Stem and Immune reference cell types are actively cycling (S, G2M stage; Fig 1C), whereas subsets of Specialised, Stem cell types are in the G0/G1 stage originating from the brain, liver, and bone marrow. We could further classify Immune cell groups into proliferating (i.e., T-cells from spleen) and quiescent (grey G0/G1 monocytes). Furthermore, both endothelial cells and hepatocytes are in the G1 stage, whereas erythroblasts are actively cycling. We next focussed on both global regulons active across multiple cell groups and cell type–specific regulons within the integrated atlas. The Irf7 (2,437 unique genes) and Runx3 (474 unique genes) are enriched in the Immune cell group (Fig 1C) (27, 28). The general TF E2F4 is enriched across most proliferating cells, while E2F7 (an atypical E2F TF) is exclusively active in a subset of highly proliferating cell groups (Fig S8C). The Foxo1 and Cebpe regulons are also enriched across multiple cell groups (Immune, Stem and Epithelial). The specific and enriched regulons include Fosb (1,352 unique genes; Endothelial and Stem), Gata2 (1,594 unique genes; Endothelial), and Gli1 (114 unique genes; Bladder cells within Specialised) (Figs 1C and S8C), Sox17 (267 unique genes; Endothelial) (29), and Cebpa (1,201 unique genes; Pancreas and myeloid single cells within Immune cell group; Fig S8C). The individual regulons and their compositions are detailed in Table S2.

The GRN inference on ~250,000 unevenly sampled single cells is computationally intensive and also impacted by scRNA-seq platform-specific biases (Figs S6A and B and S7). To address this, we generated pseudobulk cells by averaging scRNA-seq expression over 50 cells. The pseudobulk approach is computationally robust and also accounts for technical differences between atlases (additional comparison below). We re-performed the SCENIC framework on pseudobulk cells across the integrated atlas, projected individual cells on UMAP based on RAS, coloured by cell groups (Fig 1D), predicted cell cycle stage (Fig 1D), and tissue of origin (Fig S8B). We expected a better separation with pseudobulk owing to reduced technical noise (50 cell average) and more robust RAS. Consistently, the cell type separation is visually refined, with a strong overlap of cell groups across different tissues (Fig S8B) and recovery of both general and specific regulons. These include Runx3 (Immune), Sox2 (752 genes; Stem and Immune) (30), Homeodomain Pitx2 (63 unique genes from the bladder, skin, and heart), Atf3, Fos (Basal), and Foxc1 (564 unique genes; Stem) (Figs 1D and S8B and D).

We next assessed regulon activities in individual cell atlases by re-performing SCENIC (regulon scoring by AUCell) and compared

with integrated mouse atlas. The UMAP embedding based on RAS distinctly separates cell groups within individual atlases, in both single- (Fig S9A top panel) and pseudobulk cells (Fig S9B top panel). The MCA dataset has the largest number of cells, increased technical noise, lower gene detection (Fig S6A bottom row), and is enriched for Immune and Stem cell groups. Consequently, the MCA single-cell UMAP partially distinguishes reference cell groups compared with other atlases (Fig S9A top right). However, the MCA pseudobulk UMAP clearly resolves cell groups, while retaining robust regulon activities (Fig S9B). Across individual atlases, we recapitulate several integrated atlas features including global and cell group specific regulons. For example, the large Irf8 (2,988 unique genes) and smaller Tcf7 regulons (25 unique genes) are both highly specific and enriched in Immune across multiple tissues in all atlases (Figs S9A and B and S10A and B). Within individual atlases, we also observe finer cell type and tissue-specific regulon activity, including Sox17 (267 genes), Sox2 (752 genes), and Pparg (584 genes) (Fig S9A). We also observe better reference cell types mixing originating from similar tissues in pseudobulk compared with single cells (Fig S10A and B). The individual regulons and mean RAS for reference cell types are reported in Table S3.

To highlight that pseudobulk robustly captures regulon activities across cell groups in comparison with single cell, we performed several quantitative and qualitative comparisons. First, we distinguish single and pseudobulk cells by principal component analysis (PCA) for each atlas, coloured by seven cell groups (Fig 1E, see the Materials and Methods section). The pseudobulk cells are better separated than single cells by PCA, but not as distinctly as with nonlinear methods (e.g., UMAP; Fig 1C and D). We next compare the distances of individual single and pseudobulk cells to cluster centres of seven reference cell groups. Globally, the pseudobulk cells have increased distance to cluster centre than single cells, indicating a more homogeneous separation and increased cell group resolution based on RAS (Fig 1E). To compare the clustering of cell groups between pseudobulk and single cells, we computed Adjusted Mutual Information (AMI) and Completeness (Fig 1F). The AMI score is a symmetric measure of the agreement between two independent clustering labels, that is, pseudobulk and single cells, given the reference cell group labels, whereas Completeness compares clustering, given a ground truth by measuring the membership of data points to the same cluster. Across both individual and integrated atlas, the AMI scores are consistently higher in pseudobulk than single cells (Fig 1F top). Notably, the MCA AMI is significantly lower than other atlases, reflecting the poorer cell group separation in single cells compared with pseudobulk (Fig S9A and B top right). We calculate Completeness measure between

cycle stages (right), and overlaid on UMAP. The pseudobulk is generated by averaging the expression of 50 cells across same tissues, using author assigned tissue and cell type labels; and performing SCENIC regulon inference. **(E)** Principal component analysis of matched single- and pseudobulk cells based on RAS across individual atlases and coloured by seven cell groups (first two columns). For each of the seven cell groups, we plot cluster centroids (column 3) and connect single- (circles) and pseudobulk (asterisk). Box plots (column 4) represent Euclidean distance of individual single- and pseudobulk cells to respective cell group centroid. **(F)** Different measures of cluster comparison (top: adjusted mutual information, bottom: completeness) between pseudobulk and single cells across integrated and individual mouse atlases, considering seven cell groups. **(G)** Distribution of Gini coefficients per regulon in pseudobulk and single cells across integrated atlas, considering all seven cell groups. The Gini coefficient is a measure of inequality, that is, whether individual regulons contribute to individual (smaller Gini) or multiple cell groups (higher Gini). The pseudobulk cells have higher Gini coefficients and tighter distributions compared with single cells, which highlights their contribution to effectively distinguish multiple cell groups. **(H)** Comparison of regulon composition between atlases (pairwise Jaccard index) considering TM-10× as reference. Each dot represents a regulon and overlap of its target genes across three atlases. The shaded area represents 95% confidence interval from the linear regression line.

pseudobulk and single cells by comparing k-means clustering (k = 7) to our reference seven cell groups across both individual and integrated atlas (Fig 1F bottom). To measure the importance of regulons in driving integrated atlas, we computed Gini coefficient for each regulon (using RAS) across pseudobulk and single cells. The Gini coefficient is a measure of equality in a given distribution, that is, whether individual regulons drive all cell groups (Gini = 0; complete equality) or multiple regulons drive most cell groups (Gini = 1; inequality). Across integrated atlas, the pseudobulk has a higher median Gini coefficient with narrow dispersion compared with single cells (Fig 1G). Notably, the single cell RAS tend to be skewed towards lower Gini coefficient, consistent with poorer separation of cell groups in lower dimensions (UMAP and PCA; Fig 1C and D), compared with pseudobulk. We observe the same trend of Gini coefficients across individual cell groups (Fig S11A). To compare and validate the clustering between integrated and individual atlases across single and pseudobulk cells, we compute Silhouette score (Fig S11B). The Silhouette score is a measure of similarity between different clustering and considers both cohesion (within clusters) and separation (distance between clusters). We observe a positive Silhouette score for both integrated and individual atlases, with higher scores in pseudobulk cells. Consistent with previous observations, the MCA pseudobulk has significantly improved clustering and Silhouette scores compared with single cells. In addition, we assess the regulon composition similarity between pseudobulk and single cells by pairwise atlas comparison and computing Jaccard similarity index (Fig 1H). The Jaccard index is strongly correlated ($R_{pearson}$ = 0.8), highlighting that target gene compositions are similar in individual atlases (individual regulon examples described in Figs 3A–D and S18–S20). Lastly, we compare RAS between single- and pseudobulk cells and observed significantly improved correlation in individual cell groups (Fig S11C).

Given the different technical differences between individual atlases (dropouts, tissues profiled, scRNA-seq protocol, sequencing depth, etc.), we also assessed whether batch effects confound RAS across mouse atlases. Although SCENIC analysis has been shown to be unaffected by batch and technical effects (10), we performed batch correction on a common tissue (spleen) profiled by both TM-10× and TM-SS2 atlases. We apply two methods "Batch-balanced KNN" (BBKNN) and "Mutual nearest neighbours correction" (MNN) (31, 32) and visualise individual cells on t-distributed stochastic neighbour embedding (tSNE). The BBKNN and MNN-correct approaches apply correction to neighbourhood graph and expression space, respectively. The batch correction had minimal impact on resolving and overlapping similar cell types between the two atlases, compared with uncorrected data (Fig S12A). Notably, the corrected batch effects were unique to each method on tSNE space. Performing SCENIC on uncorrected and two batch-corrected datasets, we find that individual regulon activities (RAS similarity) and regulon compositions (Jaccard coefficient) are highly correlated, indicating that batch effects have little effect on regulon activity (Fig S12B and C). In summary, the pseudobulk approach accounts for technical and batch effects, robustly reports on regulon activities, and leads to better classification of cell groups across individual and integrated atlas.

For an unbiased identification of concerted regulon activity across integrated atlas, we perform cell-to-cell correlation on RAS (Fig 2A). We observe three major clusters with the largest cluster 1 composed of Immune and Stem cell groups from all atlases (Fig 1C and

D). The cluster 2 is composed of Epithelial and Stem cell group, exclusively from MCA dataset, with several sub-clusters within. The distinct MCA sub-clustering is expected, owing to increased sampling of tissues and single cells (150,889 MCA versus 93,753 TM; Fig S6A and C). The third cluster is composed of Stem and Specialised cell groups from all atlases. We observe several smaller clusters composed of individual cell types highlighting their distinct classification based on specific regulon activity (Fig 2A). Next, we performed cell-to-cell correlation within individual atlases to identify clusters composed of shared and individual cell groups, highlighting the diversity of cell types captured within each atlas. Consistent with integrated atlas, the shared clusters include "Stem and Specialised," "Immune and Stem," and "Basal and Endothelial" and are quite distinct from individual cell group clusters (Immune, Stem, Epithelial, Basal, etc.) in each atlas (Fig S13A). To further investigate shared regulatory activity across individual atlases, we performed pairwise comparison and observed strong correlation between both shared and individual cell groups (Fig S13B). In summary, the shared and individual cell group clusters validate that regulon activities correspond to true regulation in matched cell types.

To highlight the regulon crosstalk and regulation across integrated cell atlas, we performed regulon-to-regulon correlation using Connection Specificity Index (CSI) Supplemental Data 1 (21, 33). The CSI is a context dependent graph metric that ranks the regulon significance based on similarity and specificity of interaction partners, thereby mitigating the effects of non-specific interactions. Correlating across integrated atlas using CSI, we identify 174 regulons across five distinct modules and sub-modules within (Fig 2B). For broad assessment of module features, we perform Gene Ontology (GO) using all genes within regulon modules and pathway analysis on regulons (Fig S14A and B). The module 1 consists of 19 regulons (7,165 genes) involved in various cellular processes including differentiation, metabolism, and signal transduction predominantly in immune pathways (Fig S14A and B). The module 2 consists of critical TFs Gata3 and Klf16 (605 genes) that regulate multitude of cell types. Module 3 is composed of 66 regulons (8,655 genes) involved in cellular differentiation, organogenesis (including Hox and AP1 family TFs), and with significant enrichment for signal transduction pathways (Fig S14B). Module 4 consisted of 74 regulons (9,286 genes) composed of core transcriptional activators with cell cycle and messenger RNA roles (E2F, SP, and IRF family TFs), across both GO and pathway analysis. Last, cluster 5 is composed of 13 regulons (3,344 genes) involved in generalised development, tissue, and cellular organisation roles. Next, we compared whether regulon modules could be distinguished based on CSI scores within individual atlases. The larger regulon modules (Module 3 and 4) are clearly separated within individual atlases, highlighting their roles in global regulation across multiple cell groups (Fig S15A–C). The smaller modules (modules 1, 2, and 5) highlight tissue-specific regulation of different cell groups in both integrated and individual atlases. For example, the module 1 regulon Mafb regulates a subset of myeloid immune cells from microglia (Fig 2B) (34), whereas the module 5 regulon Sox2 regulates Stem and Immune group (Figs 2B and S9A).

To investigate regulon crosstalk within and between modules across the integrated atlas, we devised an undirected regulon network considering the most interacting regulons with stringent CSI association (CSI > 0.7; Fig 2C). As expected from CSI correlation matrix (Fig 2B), the regulons

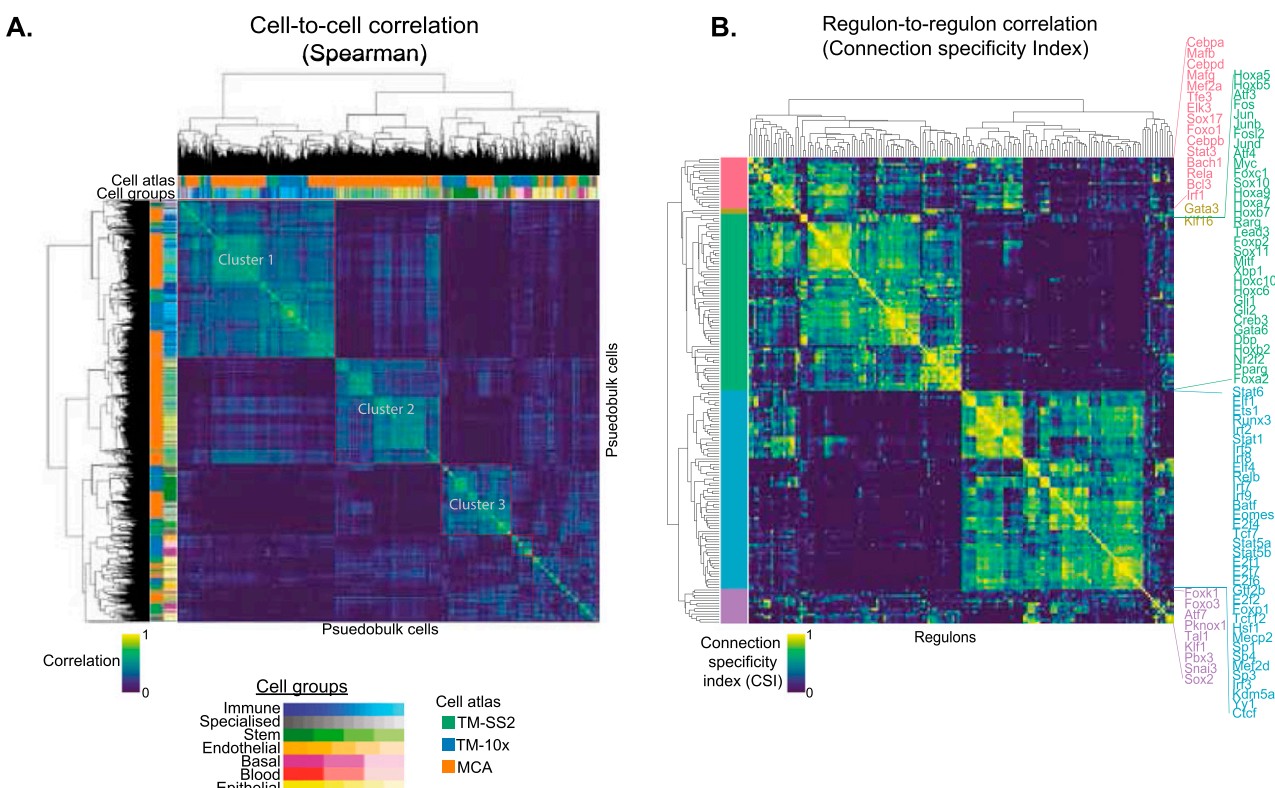

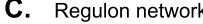

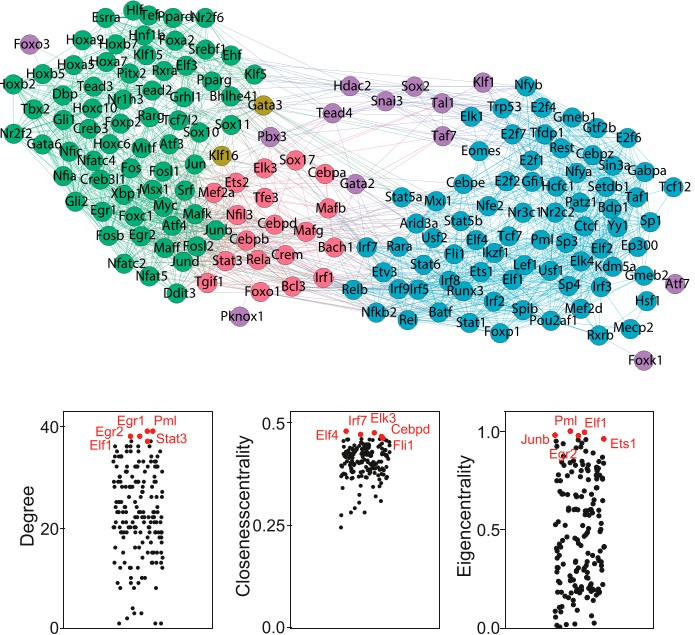

**Figure 2. Regulatory network uncovers broad acting and cell-type specific regulon activites. (A)** Spearman cell-to-cell correlation map across three atlases. The first column (and topmost row) indicates the respective mouse atlases, whereas the second column (and second row) indicates the 55 reference cell types. The clusters are highlighted in red rectangles. **(B)** Connection Specificity Index (CSI) matrix highlights regulon-to-regulon correlation in pseudo-bulk cells across integrated atlas. Hierarchical clustering of regulons identifies five distinct regulon modules (first column), which capture both global and distinct regulatory roles across cell groups and tissues. Selected regulons are coloured by module and listed next to heat map. **(C)** Undirected regulon network generated from strongly correlated CSI scores (Fig 2B). Each regulon is represented as a node, and regulons pairs with strongest associated interactions (CSI scores > 0.7) are connected with undirected and unweighted edges. The larger modules 3 (green) and 4 (blue) are bridged by smaller modules. Bottom: examples of individual regulons contributing to different network features (degree, closeness centrality, and eigen centrality).

within modules have higher connections than across modules implying concerted regulation in cell types across integrated atlas. We also assess several network features to determine regulon importance for individual modules as well as regulon network. Notably, the smaller modules (1, 2, and 5) bridge the nodes between larger modules (3 and 4) within the network. Within individual atlases, we find that the global regulon network is largely retained irrespective of regulon composition differences within atlas (Fig S15D–F). We highlight regulons with important regulatory roles in reference cell types within individual atlases (Fig S16). Assessing the different network features across the integrated network, we find Cebpd (module 1), Gata3 (module 2), and Hdac2 (module 5) are the key bridge nodes (betweenness centrality) traversing the shortest path through the network. The top intra- and inter-module regulons have highly correlated network features (degree, closeness, and Eigen centrality with regulon composition; Fig 2C bottom). The network features across integrated atlas are detailed in Table S4.

To further validate the modules across integrated regulon network, we perform several in silico comparisons. First, our framework includes RCisTarget (as a part of SCENIC) for defining regulons, that is, TFs and direct target genes. RCisTarget cross-matches identified regulons with known and annotated TF target databases, prunes indirect co-expressed targets, and enables scoring of TF–TF and TF-target relationships. Consequently, all direct targets of a given regulon harbour the regulon motif at respective promoters. In addition, we expect and observe many regulons within individual modules to share overlapping motifs (motif correlation in Fig S17A). We also report a few representative examples of regulons and their motifs within individual modules (Fig S17A). We next assessed whether regulons crosstalk across the integrated network are mediated through protein–protein interactions (PPi). Comparing and overlaying the annotated PPi from STRING (35), we validate 57% of regulon network connections (Fig S17B). Since each STRING annotated PPi is assigned a combined score (measure of confidence), we compared our regulon network with the STRING combined score (in 20% bins; Fig S17C). Consistently, the regulon network connections have the highest STRING combined score. In addition, we also observe a strong positive relationship between regulon CSI and STRING combined score, validating the regulon network interactions from experimental evidence (Fig S17D and Table S5). Last, we compared our regulon network for essential genes in the Online Gene Essentiality database (OGEE) (36). We observe 109 essential genes (70%) in our regulon network with strong representation across all modules (Fig S17E and Table S5), further highlighting the regulon importance across integrated network.

Next, we focussed on regulons with differential composition that drive individual cell types (Figs 3A and S16). The regulon Cebpe consists of 1,342 unique genes (TM-10×: 332, TM-SS2: 531, and MCA: 479 genes) with 189 common and direct targets. The Cebpe activity is highly specified in granulocyte and monocytes, consistent with its known role in lineage determination (Fig 3A) (37). The Irf8 is a master regulator of monocytes and dendritic cells and is important for both adaptive and innate immunity (38). We observed 641 shared targets and specific activity in monocytes and macrophages (Fig 3B). We find that regulons with few shared direct targets across cell atlases have specific and consistent activity. The Lef1 and Hoxb7 regulons have fewer overall targets genes, only five shared targets between cell atlases, but with specific activity in T-cells (39) and kidney epithelial

cells (40), respectively. Several global and cell type–specific regulons with differential compositions are presented in Figs S18–S20.

To further validate our regulatory framework for atlas-scale analysis, we performed GRN inference using an alternative method "bigSCale2" considering TM-10× atlas (41). The "bigSCale2" approach uses expression correlation to calculate regulatory network and does not distinguish between direct and indirect TF-targets. Comparing the two methods, we find 117 regulons (67%) co-identified by both methods, whereas 57 regulons (33% and direct targets within) exclusively captured in our SCENIC framework (Fig S21A). Computing the Jaccard index, we find only 95 regulons with composition similarity between both GRN inference methods (Fig S21B). In summary, the SCENIC framework robustly identifies regulons and their direct targets for atlas-scale analysis. We also compared GRN scoring between SCENIC (AUCell) and an alternative approach (VIPER), which computes a normalised enrichment score (NES) per regulon, for a defined cell type (or group) by comparing against other defined cell types (42). We consider regulons (GRNBoost and RcisTarget) across B cells from the TM-10× atlas for comparison (Fig S22A and B). Although direct comparison between the two approaches is tricky because of underlying scoring methods, we plot the regulon correlation between VIPER enrichment scores and mean RAS (NES versus mean AUC; Fig S22A). However, the correlation is significantly improved when considering the VIPER enrichment scores (42) with regulon specificity scores (21), indicating specific cell type enrichment (Fig S22B).

Last, we assess the functional importance of regulon activity by investigating mixed-lineage transitions during myeloid cell-fate determination using scRNA-seq (43). The Irf8 regulon and its regulatory interactions are critical for monopoiesis and have a reciprocal dynamics with Gfi1-driven granulocyte specification (43). We analyse granulocytic and monocytic specification in wild-type and Irf8$^{-/-}$ progenitors using scRNA-seq data (Fig S23A), infer regulons, and score regulon activity in single cells (Fig S23B). Both scRNA-seq expression counts and regulon activities separate different cell types and capture the shift in Irf8$^{-/-}$ cells towards granulocyte lineage (Fig S23A–D). Comparing the Irf8 regulon across monocytes, granulocytes, and Irf8$^{-/-}$ cells, we observe preferentially high composition of direct targets in Monocytes (542 genes) over granulocytes (148 genes), consistent with cell-fate roles (Fig S23E) (43). Notably, the Irf8 regulon is significantly perturbed both in composition (direct target genes) and activity (target regulation) across Irf8$^{-/-}$ cells (Fig S23D and E), highlighting the functional importance of regulon activity in mediating cell states and cell types.

## Discussion

As major tissue, organ, and organism expression atlases are increasingly generated (44, 45), it is critical to also decipher mechanistic gene regulatory programs for refining functional cell state and cell type definitions. Here, we highlight a computational approach to infer regulons and link their specific cell type activity with functional roles, from integrated scRNA-seq cell atlases. To resolve the author assigned cell type labels across cell atlases, we standardise and categorise single cells into broadly defined seven cell groups and reference cell types. In our study, we project and map cell groups across different cell atlases, which indeed diminishes the

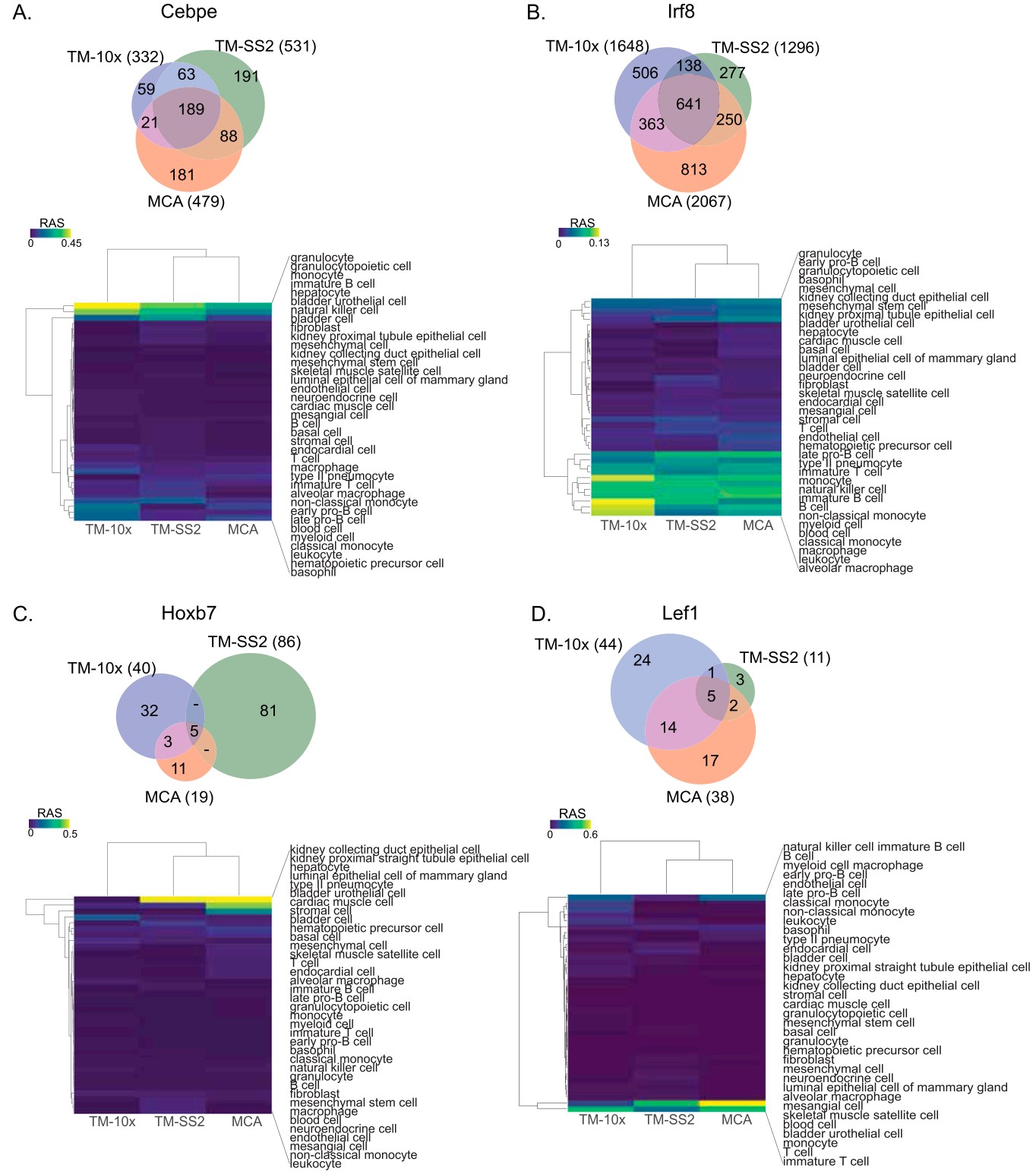

**Figure 3. Regulon compositions and activities across atlaseslases. (A, B, C, D)** Venn plots of representative individual regulons, gene compositions, overlap across individual atlas and specific cell type regulation (A) Irf8, (B) Irf8, (C) Hoxb7, and (B) Lef1. The heat map represent z-scaled mean regulon activity score across different cell types.

resolution of individual cell types. However, it allows us to converge and group similar cell types together irrespective of differences in cell atlases including tissue sampling, scRNA-seq platform, and sequencing depth.

For regulon identification, inference, and scoring, we used GRNBoost, RcisTarget and AUCell (as in reference 10); however, alternative inference methods have been proposed with improvements for both directed and

undirected networks (9, 12, 46, 47). A fundamental caveat of recent GRN methods is the requirement of a priori pseudotime or cellular trajectory, which makes them incompatible for atlas-scale analysis. Our integrative analysis on three atlases uncovers global regulon modules that operate on multiple cells types, as well as specialised regulons critical for cell type definition and identity. Through a variety of in silico comparisons, we highlight the robustness of pseudobulk cells in effectively classifying cell groups and highlight regulatory crosstalk. The global regulon network is recapitulated in individual cell atlases using both single- and pseudobulk cells, validating the regulatory crosstalk in individual cell groups. The functional consequence of regulon composition and activity is highlighted during the lineage transition from monocytes to granulocytes in Irf8k cells. Our integrated computational atlas with standardised classification of cell groups, global, and cell type–specific regulons across three MCAs presents a valuable resource for the single-cell community.

# Materials and Methods

## Datasets

### Tabula Muris

The TM scRNA-seq dataset contains single cells profiled using both 3′end 10× Chromium and full-length Smart-Seq2 (17). The data were retrieved through the data portal (https://figshare.com/projects/Tabula_Muris_Transcriptomic_characterization_of_20_organs_and_tissues_from_Mus_musculus_at_single_cell_resolution/27733). The data contained 1,23,878 single cells. The Smart-seq2 dataset consists of 53,760 single cells from 18 tissues classified into 81 cell types, whereas the 10× Chromium contains 70,118 single cells from 12 tissues classified into 55 cell types. After filtering the non-annotated cell types, we obtained 44,779 and 54,865 single cells from Smart-seq2 and 10× Chromium, respectively. The annotated cell types from Smart-seq2 and 10× Chromium are referred to as author assigned cell type labels.

### Mouse cell atlas

The MCA (18) scRNA-seq dataset contains single cells profiled using authors 3′ end microwell method. The data were retrieved through the data portal (https://figshare.com/articles/MCA_DGE_Data/5435866). After filtering non-annotated cells types, we obtained 2,33,994 single cells from 38 tissues classified into 760 cell types. The annotated cell types from across 3′ end microwell method are referred to as author assigned cell type labels.

### Myeloid differentiation

The myeloid differentiation dataset contains 382 wild-type (9 cell types) and 62 Irf8$^{-/-}$ cells (43). The scRNA-seq expression matrices were retrieved from data portal (https://www.dropbox.com/sh/yqlclftyolwqy7y/AADVD-_IOqpXQx8PlWcywMypa?dl=0) (48).

## Data processing

### Data normalization and scaling

We use Scanpy (version 1.4) for normalization of all datasets (49) using the pre-processing functions for cell library size (scanpy.pp.normalize_per_cell) and log-transformation (scanpy.pp.log1p). We regress the variance arising from variable library size and mitochondrial gene count fraction, and scale genes (zero mean and unit variance) using in-built functions (scanpy.pp.regress_out and scanpy.pp.scale, respectively). The Highly Variable Genes for each dataset are calculated using in-built functions (scanpy.pp.highly_variable_genes) with default parameters.

### Pseudobulk

For creating pseudobulk cells, we randomly sampled 50 cells from author assigned reference cell type within a given tissue. Only genes with non-zero counts are used for averaging. This approach potentially removes author assigned cell types consisting of fewer than 50 cells (very rare cells).

### Cell cycle stage prediction

The cell cycle stage prediction is performed using Scanpy function (scanpy.tl.score_genes_cell_cycle) to score S and G2M-specific genes. Each single cell has an S- and G2M-score and is assigned, respectively, based on the highest scoring class. If neither the S-score nor the G2M-score exceeds 0.5, the cells are assigned as G1 phase. The reference cell cycle phase marker genes (50) used for scoring can be found here (https://github.com/theislab/scanpy_usage/blob/master/180209_cell_cycle/data/regev_lab_cell_cycle_genes.txt).

## Mapping author-assigned cell type labels to common reference

### Reference cell types

We first devise a common reference for mapping different author assigned cell type labels. We choose TM 10× cell type labels as reference cell types as it has the fewest annotated cell types for effective integration. The reciprocal reference using either TM Smart-seq2 or MCA lead to unresolved and undefined cell types and poor mapping. We manually curated the reference cell types to seven cell groups (Fig S3B).

### scMAP

We map both TM Smart-seq2 and MCA to TM 10× separately using scMAP (version 1.4.1) with default parameters (23). We use the function "selectFeatures" for identifying features and use the common feature set (Intersection) for mapping. This further reduces the contribution of cell types either identified in single atlas or without any common features with reference cell types. For example, none of the MCA single cells mapped to reference cell type "Keratinocytes" in TM 10×. Similarly, none of the TM Smart-seq2 single cells mapped to reference cell type "Duct epithelial cells" in TM 10×. In the last step, we further exclude non-mapping cells. The remaining single cells from TM 10× (54,865 cells), Smart-seq2 (38,888 cells), and MCA (150,889 cells) are used for regulon inference.

## Inferring GRNs

### Feature selection for pySCENIC

To retain a large but stringent feature size while accounting for technical atlas differences, we select the features that are expressed in 10% of pseudobulk cells for downstream analysis (Fig 1B). Similarly, we select genes expressed in 10% wild-type cells (1,002 genes) from the myeloid differentiation dataset (43).

### Dataset pre-processing for pySCENIC

The raw datasets are normalised using Scanpy pre-processing functions for cell library size (scanpy.pp.normalize_per_cell) and log-transformed (scanpy.pp.log1p). No additional scaling of genes was performed.

### Running pySCENIC

We implement the three steps for pySCENIC pipeline (10). First, GRNboost is run on filtered expression matrix using list of TFs (https://resources.aertslab.org/cistarget/motif2tf/motifs-v9-nr.mgi-m0.001-o0.0.tbl). Second, RcisTarget is used to infer direct targets using "mm9-500bp-upstream-7species" and "mm9-tss-centred-10kb-7species" (https://resources.aertslab.org/cistarget/). The defined regulons are TFs and their direct target genes harbouring significant TF motif enrichment. Third, RAS is calculated using AUCell as the area under the recovery curve of all genes identified within the regulon. All the steps are run with default parameters. The regulon inference identifies 233 regulons in TM 10× (median composition of 141.5 genes), 222 regulons in TM Smart-seq2 (median composition of 195 genes), and 222 regulons in MCA (median composition of 151 genes).

Similarly, we identify 154 regulons (median composition 93.5 genes) from wild-type cells in myeloid differentiation dataset (43). We also separately ran pySCENIC pipeline on Monocytes (191 regulons, median 48 genes), granulocytes (181 regulons, median 54.5 genes), and Irf8$^{-/-}$ cells (136 regulons, median 69.5 genes), respectively. To specifically infer Irf8 regulon activity (Fig S22D) in both wild-type and Irf8$^{-/-}$ cells, we repeated AUCell 50 times and used the averaged activity score.

## Cell type similarity based on regulon activity

### Spearman correlation

We calculated pseudobulk cell-to-cell spearman correlation coefficients based on RAS to quantify cell type similarity using "scipy.stats.spearmanr" (version 1.1.0). The pseudobulk spearman correlation coefficients are classified by hierarchical clustering using "seaborn.clustermap" function (version 0.9.0) with default parameters. The force directed graphs only link edges where the spearman correlation coefficients are greater than 0.5.

## Embedding

### PCA

PCA is performed on RAS using "scanpy.tl.pca" with default parameters.

### UMAP

We performed Uniform Manifold Approximation and Projection (24) using the Scanpy function "scanpy.tl.umap" with default parameters.

## Comparison of RAS and regulon composition between single- and pseudobulk cells

### PCA and cluster centres

To compare the RAS between single- and pseudobulk bulks, we first plotted pseudobulk cells on PCA (sklearn.decompositin.pca) and projected the single cells onto the same embedding. For individual single- and pseudobulk cell, we calculated the Euclidean distances to cell group centres.

### AMI and completeness

For clustering comparison between single- and pseudobulk cells, we performed K-means clustering (using k = 7) and compared clusters to ground truth, that is, seven reference cell groups. The AMI (sklearn.metrics.ajusted_mutual_information_score) and completeness (sklearn.metrics.completeness_score) is calculated on RAS of individual regulons from single- and pseudobulk cells. Similarly, the RAS correlation is quantified between single- and pseudobulk cells for global and individual cell groups.

### Gini coefficient

To measure equality of RAS in classifying individual and global cell groups, we calculate Gini coefficient of RAS per regulon between both single- and pseudobulk cells.

$$G = \frac{\sum_{i=1}^{n}(2i - n - 1)x_i}{n\sum_{i=1}^{n}x_i}$$

## Comparison between integrated and individual mouse atlases

For clustering comparison between integrated and individual atlas, we performed k-means clustering (using k = 7) and calculate silhouette score ("sklearn.metrics.silhouette_score"), by comparing with ground truth, that is, seven reference cell groups.

## Regulon modules and regulon networks

### Connection specificity modules and network

The CSI is calculated for each pair of regulon (from Pearson correlation coefficient) and is a measure to identify regulatory partners (21, 33).

The CSI for two nodes A and B is calculated by:

$$CSI_{AB} = 1 - \frac{No. \ of \ nodes \ connected \ to \ A \ or \ B \ with \ PCC \geq PCC_{AB} - 0.05}{n_y}$$

Where the Pearson correlation coefficient (PCC) is the interactional correlation between A and B.

To identify regulon modules (Fig 1F), we use hierarchical clustering (scipy.cluster.hierarchy.fcluster) on regulon linkage matrix (scipy.cluster.hierarchy.linkage) using method "average" to calculate Euclidean distances between clusters. The Pairwise distance between regulons is calculated by "scipy.spatial.distance.pdist" with Euclidean metric. We filter select regulon modules, that is, co-active regulons (regulons pairs) with CSI greater than 0.7 and project on a force directed graph, coloured by regulon modules (Fig 1G).

## Functional analysis of regulon modules

### GO

We use ClusterProfiler (v3.10.1) for GO analysis (Biological Processes) (51). For significant GO terms within the regulon module, we use enrichGO function considering all unique genes within the regulon module as gene set. The GO comparison across modules was performed using compareCluster function. Significant

terms are selected using *P*-value cutoff (*P* < 0.05) after adjusting for multiple testing using Benjamini–Hochberg correction.

### Pathway analysis

The regulons within individual modules are directly used for pathway analysis using Reactome (https://reactome.org/PathwayBrowser/) with default parameters (52). For each module, we quantify the significantly enriched terms (*P*-value < 0.05) within each high-level pathway term (e.g., Immune System, Metabolism, Developmental Biology etc.). The terms and relationships were downloaded from Reactome directly (pathways: https://reactome.org/download/current/ReactomePathways.txt and relationships: https://reactome.org/download/current/ReactomePathwaysRelation.txt).

## Batch effect correction

We applied two different batch correction methods on TM-10× and TM-SS2 atlases. We used pseudobulk cells from the spleen, which was profiled by both atlases.

### MNN-correct

We corrected the expression space of the two atlases using the Scanpy implementation of MNN-correct ("scanpy.pp.mnn_correct") with parameters (svd_dim = 5 and k = 10), using the two atlases as batch key.

### BBKNN

We corrected the neighbourhood graph of the two atlases also using the Scanpy implementation ("bbknn" library version 1.3.1, with parameters (neighbors_within_batch=10, n_pcs=10, trim=50), using the two atlases as batch key).

### Non-corrected

The non-corrected expression space of the two atlases was created by concatenating the individual Scanpy AnnData objects ('scanpy.AnnData.concatenate' with join='inner').

For both the non-corrected and batch corrected data, we compute regulons using pySCENIC CLI that includes RCisTarget (database: "mm9-tss-centred-10kb-7species") for cross matching and regulon pruning. For regulons identified in both non-corrected and batch corrected data, we compute the spearman correlation of RAS between datasets. For each regulon predicted in the batch correction datasets, we compute Jaccard index (sklearn.metrics.jaccard_similarity_score) as a measure of composition similarity to the non-corrected dataset.

$$J(A, B) = \frac{|A \cap B|}{|A \cup B|} = \frac{|A \cap B|}{|A| + |B| - |A \cap B|}$$

## Network validation

### STRING

The experimental annotated and scored PPi were downloaded from the static STRING database (https://stringdb-static.org/download/protein.links.v11.0.txt.gz) alongside their "Combined score," which is a measure of confidence of interaction. We classified the PPi and CSI in 20 and 10 percentile bins, respectively, based on "Combined score," and compared the regulon network node-edges pairs.

### OGEE

The OGEE gene essentiality table was retrieved from http://ogee.medgenius.info/file_download/gene_essentiality.txt.gz, and gene identifer to name mapping was performed using http://ogee.medgenius.info/file_download/genes.txt.gz. We only considered mouse genes for analysis.

## Regulon importance and integrated network features

Having constructed the integrated regulon network, the network features ("Degree," "Closeness centrality" and "Eigen centrality") are calculated using Gephi (0.9.2) with default parameters.

## Comparison of regulon motifs

For each individual regulon across the integrated network, we obtained the TF binding motif from JASPAR (53). We used a published database containing Pearson correlations between TFs position weight matrix (54, 55) and subset, visualised the TFs from integrated regulon network.

## Comparison with alternative GRN (regulon) scoring method

We used VIPER (version 1.16) (42), as an alternative to scoring regulons with AUCell. To facilitate comparison, we used the regulons inferred by GRNBoost and RCisTarget on B-cells from the TM-10× cell atlas (222 regulons, 10,119 targets and 78,742 interactions), applied VIPER ("rowTtest") to get B-cell signatures, and compared with other cell types (z-score). The *t* test null model was made with 20 permutations and reposition, and the NES are computed using "msviper" function. We compare VIPER NES with regulon specificity score as described in reference 12.

## Comparison with alternative GRN inference method

We compared our atlas-scale GRN inference using SCENIC with an alternative published GRN method "bigSCale" (41) on TM-10× atlas considering the same gene-set of 11,245 genes. We used the most recent bigSCale version 2 "compute.network" with parameters (speed.preset='fast' and clustering='direct') and default Pearson correlation cutoff (R = 0.9) for network construction. bigSCale2 captures 117 out of 174 regulons from our SCENIC consensus network. Using Jaccard index (sklearn.metrics.jaccard_similarity_score), we also quantify the regulon composition overlap between bigSCale2 and our SCENIC consensus network.

## Computational infrastructure

The computational analysis was performed on DeIC National High-Performance Computing cluster (ABACUS 2.0) with each node consisting of two Intel E5-2680v3 CPUs with each 12 cores and with

64 or 512 GB RAM. Dask was used to parallelize compute intensive processes across several nodes (56, 57, 58).

## Data Availability

The supplementary document contains the full data sources analysed in the current study. The Jupyter notebooks detailing all the analysis steps can be found here: https://github.com/Natarajanlab/Single-cell-regulatory-network.

## Supplementary Information

## Acknowledgements

The authors thank Lars Grøntved and other Functional Genomics Unit members for helpful discussions and comments on the manuscript. The authors also thank DeIC National HPC Centre (ABACUS 2.0) at SDU, Denmark, for computational resources. Funding: AF Møller and KN Natarajan acknowledge the Sino-Danish Center and Louis Hansen Foundation (J.nr.20-2B-6705) for funding support. The research in KN Natarajan lab is supported by Danish Institute of Advanced Study, Villum Young Investigator grant (VYI#25397) and Novo Nordisk grants (#NNF18OC0052874 and #NNF19OC0056962).

### Author Contributions

AF Møller: data curation, formal analysis, investigation, visualization, and methodology.
KN Natarajan: conceptualization, formal analysis, supervision, funding acquisition, validation, investigation, visualization, methodology, project administration, and writing—original draft, review, and editing.

### Conflict of Interest Statement

The authors declare that they have no conflict of interest.

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
