## [Reviewer comments · Life Science Alliance]

Life Science Alliance

Predicting gene regulatory networks from cell atlases

Andreas Møller and Kedar Natarajan

DOI: <https://doi.org/10.26508/lsa.202000658>

Corresponding author(s): Kedar Natarajan, University of Southern Denmark

Review Timeline:

Submission Date:	2020-01-23
Editorial Decision:	2020-03-18
Revision Received:	2020-06-15
Editorial Decision:	2020-08-10
Revision Received:	2020-08-24
Accepted:	2020-08-31

Scientific Editor: Shachi Bhatt

Transaction Report:

March 18, 2020

Re: Life Science Alliance manuscript #LSA-2020-00658

Dr. Kedar Nath Natarajan
University of Southern Denmark
Biochemistry and Molecular Biology
Campusvej 55
Odense 5230
Denmark

Dear Dr. Natarajan,

Thank you for submitting your manuscript entitled "Predicting gene regulatory networks from cell atlases" to Life Science Alliance. The manuscript was assessed by expert reviewers, whose comments are appended to this letter.

As you will see, the reviewers point out several weaknesses of your analyses that would need addressing to allow publication here. Should you be prepared to address the concerns raised, we would be happy to invite you to submit such a revised version to us. But please consider your options carefully, we will need strong support from both reviewers on a revised version in order to move forward here. Should you decide to embark into the revision, it would be important to address the following reviewer concerns convincingly:

- the code must be made available to the reviewers (we also mandate availability to all upon publication)
- provide quantifications that are currently missing
- more comparisons are needed to demonstrate the relevance of the approach and the definition of matching cell types needs further support via orthogonal approaches
- consider computational validation of the regulator-target/network relationships
- provide all information necessary for proper re-review

The typical timeframe for revisions is three months. Please note that papers are generally considered through only one revision cycle, so strong support from the referees on the revised version is needed for acceptance. We are aware that many laboratories cannot function fully during the current COVID-19/SARS-CoV-2 epidemic and therefore encourage you to take the time

necessary to revise the manuscript to the extent requested above. We will extend our 'scoping protection policy' to the full revision period required. If you do see another paper with related content published elsewhere, nonetheless contact me immediately so that we can discuss the best way to proceed.

Thank you for this interesting contribution to Life Science Alliance. We are looking forward to receiving your revised manuscript.

Sincerely,

B. MANUSCRIPT ORGANIZATION AND FORMATTING:

We encourage our authors to provide original source data, particularly uncropped/-processed electrophoretic blots and spreadsheets for the main figures of the manuscript. If you would like to add source data, we would welcome one PDF/Excel-file per figure for this information. These files

will be linked online as supplementary "Source Data" files.

Reviewer #1 (Comments to the Authors (Required)):

Moller and Natarajan infer mouse-specific regulatory networks from scRNA-seq data. For this, the authors interrogated three publicly available mouse cell atlases from "Tabular Muris" and "Mouse Cell Atlas", that were profiled by different scRNA-seq technologies and come with different numbers of cells, tissues, and annotated cell types. In order to make the atlases comparable, they were integrated by mapping all single-cells to 7 "cell groups" that cover 55 "reference cell types" across various tissues.

The authors used the SCENIC workflow to infer the gene regulatory networks and predicted their activity, measured as RAS (Regulon Activity Score) using the statistical method AUCell.

Using the TF activity matrix for dimensionality reduction via UMAP revealed a good separation of distinct cell-groups. Contextualizing the UMAP plot with regulon activities revealed cell-type specific and consensually active regulons.

The authors performed simple yet effective cell-to-cell and regulon-to-regulon correlation to identify cells with similar gene regulation profiles and regulons with similar activity. For the latter, the authors interpreted the results as potential TF crosstalk and found in total 5 modules that were subsequently characterized via GO and pathway analysis.

Finally, the authors demonstrate the usage of GRN inference and regulon activities highlighting the importance of *Irf8* during lineage development.

It appears that the study was carefully and overall well performed, although without having access to the code this can not be fully confirmed.

While the approach is not particularly novel, the integrated GRN (as well as the individual GRN) might be of interest to those in the community that do not have the capability or skills to run the tools on their own.

After revising the manuscript, some questions still remain open and some statements could be supported by further analyses. Specific comments are provided hereafter:

Major comments

1) As far as we understand, the authors integrated all three atlases via matching of cell types, and considering the common core of genes (11425 genes) but inferred GRN for each individual atlas. ("... we identify 279 regulons with >60% shared between cell atlases..."; page 6). For an integrated analysis, we would expect that the authors infer a single consensus GRN using all 3 integrated cell atlases.

On page 8 2nd paragraph the authors state: "Next we focus on individual cell atlases, re-performing GRN framework...".

For us, it is not entirely clear where the difference between the "integrated" and "individual cell atlas" GRNs is. If the only difference is the consideration of all genes of the respective atlas it does not surprise me that the authors find on page 8: "The regulon activities are highly consistent between integrated and individual cell atlases...", as more than 80% of the genes are the same.

2) The authors state in the introduction that there are multiple methods to infer gene regulatory networks. We missed the motivation why the authors chose SCENIC.

3) AUCell results lack directionality. There are other statistical approaches to analyse regulons that provide signed TF activities. Authors could compare to these, or at least elaborate on whether this could be relevant for their study.

4) The authors state: "The cell-type separation was refined with pseudo-bulk cells and we robustly recover both general and specific regulons." - Given Figure 1 C and D, we are not entirely sure whether this statement is actually true. we would like to see a quantitative analysis to prove that using pseudobulk actually improved cell type separation.

5) The authors state: "The regulon activities are highly consistent between integrated and individual cell atlases, across single and pseudobulk cells (Supplementary Fig 8A-G, S9A-B)." - The authors support their statement with individual examples but also here we would like to see a comprehensively quantitative analysis.

6) For the sake of transparency and reproducibility, the authors should make their analysis code publicly available. (e.g. via GitHub). They mention a GitHub repository in the paper, but no link is provided?

Minor comments:

1) In addition to the review of GRN reconstruction by Fiers et al., also a recently published benchmark of those methods could be referred to - Pratapa et al., Nature Methods (<https://doi.org/10.1038/s41592-019-0690-6>)

2) Typo in Supplementary Figure 5 A and B: altlast -> at least

3) Related to Figure 1C and others: Even though the term RAS (regulon activity score) is defined in the main text and method section, it would be helpful to define this term also in the legend/caption.

54 "The individual regulons, their compositions and activity scores are detailed in Supplementary Table 1" page 7 - This is a wrong reference, as Supplementary Table 1 contains the results from scMAP. Also the activity scores are not reported but the module number/ID.

5) Why is the pathway analysis (Supplementary Figure 11 B) performed on regulons and not on all genes within regulons as in the GO analysis?

6) No logical order supplementary Figures. Supplementary Figure 3 B is referenced first (page 5)

Reviewer #3 (Comments to the Authors (Required)):

In this paper, the authors use existing cell atlases in mouse to build regulatory networks. They do this by first applying scMap to identify similar cell populations in each atlas and then apply SCENIC to do network inference. Network inference is done on the entire merged dataset, on a downsampled version and also on each cell atlas. Results are compared based on the consistency of recovery of regulons, defined by transcription factors enriched in a set of co-regulated/co-expressed genes. Although the inference of regulatory networks and modules from these published compendia is interesting to the community, and the finding that cell types could be discriminated by the regulon community is insightful, the presented approach and analysis does not seem sound and there are several points that I think need additional explanation or analysis.

1. Definition of cell types. The authors don't really provide any detail about how they determined the generalized vocabulary of 55 cell types and species. They simply cite Supp Fig 3A and it is unclear how to understand the process from this figure.

2. The authors claim that they are able to robustly map author specified cell types, but I am finding the figures hard to read; there is too much cross-edges between the cell types across the different datasets. It might be good to quantify this.

3. Although the authors say they are able to define matching cell types, I feel that using one or two additional approaches to correct for batches could be beneficial. E.g. conos, scanorama, liger, seurat are pretty standard and new approaches that people have applied and compared and should be used to verify their results.

4. The definition of a regulon and its comparison across datasets needs to be more precise. Is a module, a set of co-expressed genes or co-expressed and co-regulated, or co-regulated? They define a regulon as "(modules of enriched TFs and direct regulators), which would suggest that a regulon is defined by group of regulators, but the downstream analysis only uses one TF at a time.

5. Furthermore, the authors use a measure "Correlation Specificity Index" to define similarity between modules to examine the similarity of modules, but this is not well-defined. They mention Pearson correlation between regulons, but the regulon is a collection of genes and TFs/regulators. Hence using some mathematical notation could be beneficial here.

6. The authors they use a "variety of different feature sets" and cite Supp Fig 5A-D. But all this shows is a set of venn diagrams and it is not clear what the criteria is for using a gene set. I also did not understand what the authors are showing in supp 5E. They say "The regulon definition was highly similar with differential gene composition, owing to variable sequencing (Supplementary Fig 5E)" I am not sure what is meant by "Variable sequencing" and I am not convinced the regulons are similar. It seems only the number of genes per regulator is plotted, but it does not inform us about the composition of the target set.

7. The claim that "groups have good separation based on regulon activity scores" needs to be better quantified. They are using the original cell groups and the cell type labels to color the cells in UMAP coordinates. This grouping could be better quantified by clustering and checking if the clusters do correspond to the cell types.

8. I did not see the relevance of the downsampled data analysis and the authors don't do a systematic comparison of whether the results are actually the same or different. They say that cell types are more refined and they again find global and specific regulons, but this is very qualitative and more principled comparisons are needed.

9. Similarly, I found the analysis of the individual atlases not as insightful. It was hard for me to see from Supp Fig 8,9 how we can infer consistency since these are different umap plots and we can really compare these projections. The cells could again be clustered and cluster-cell type association could be established and the regulons could be compared thereafter, or even, without clustering.

10. The *lrf8* mutant versus wild type analysis again seems disconnected and does not naturally follow from the cell-atlas regulon analysis. *lrf8* was one of regulators, but there were several others that were discussed. Furthermore, the targets of *lrf8* inferred in the cell-atlas were not actually validated. Rather, the new scRNA-seq dataset was used to redefine modules and *lrf8* was found as a regulator here.

11. In general, there is no computational or experimental validation of the regulator-target/network relationships. It is not clear how accurate the inferred networks are. Additional comparison to existing databases of TF-target relationships is needed to support the inferred GRNs.

Minor:

Euclidian should be Euclidean.

BLACK = reviewers comments

BLUE = our response

RED = revised text

Reviewer #1 (Comments to the Authors):

Moller and Natarajan infer mouse-specific regulatory networks from scRNA-seq data. For this, the authors interrogated three publicly available mouse cell atlases from "Tabular Muris" and "Mouse Cell Atlas", that were profiled by different scRNA-seq technologies and come with different numbers of cells, tissues, and annotated cell types. In order to make the atlases comparable, they were integrated by mapping all single-cells to 7 "cell groups" that cover 55 "reference cell types" across various tissues.

The authors used the SCENIC workflow to infer the gene regulatory networks and predicted their activity, measured as RAS (Regulon Activity Score) using the statistical method AUCell. Using the TF activity matrix for dimensionality reduction via UMAP revealed a good separation of distinct cell-groups. Contextualizing the UMAP plot with regulon activities revealed cell-type specific and consensually active regulons. The authors performed simple yet effective cell-to-cell and regulon-to-regulon correlation to identify cells with similar gene regulation profiles and regulons with similar activity. For the latter, the authors interpreted the results as potential TF crosstalk and found in total 5 modules that were subsequently characterized via GO and pathway analysis.

Finally, the authors demonstrate the usage of GRN inference and regulon activities highlighting the importance of Irf8 during lineage development.

It appears that the study was carefully and overall well performed, although without having access to the code this can not be fully confirmed. While the approach is not particularly novel, the integrated GRN (as well as the individual GRN) might be of interest to those in the community that do not have the capability or skills to run the tools on their own.

After revising the manuscript, some questions still remain open and some statements could be supported by further analyses. Specific comments are provided hereafter:

We thank the reviewer for his/her comments, acknowledging the simplicity, effectiveness, design of our study in characterising atlas-scale regulatory network and its usefulness to the community.

We would like to emphasize that our motivation is to extract and validate regulatory information by integrating atlas-scale datasets, especially as various cell, tissue and organism level atlases are being increasingly generated.

We apologise for not enabling the Github link with jupyter notebooks (incl. revision analysis), which can be found here: <https://github.com/Natarajanlab/Single-cell-regulatory-network>.

Major comments

R1.1: As far as we understand, the authors integrated all three atlases via matching of cell types, and considering the common core of genes (11425 genes) but inferred GRN for each individual atlas. ("... we identify 279 regulons with >60% shared between cell atlases..."; page 6). For an integrated analysis, we would expect that the authors infer a single consensus GRN using all 3 integrated cell atlases.

On page 8 2nd paragraph the authors state: "Next we focus on individual cell atlases, re-performing GRN framework...".

For us, it is not entirely clear where the difference between the "integrated" and "individual cell atlas" GRNs is. If the only difference is the consideration of all genes of the respective atlas it does

not surprise me that the authors find on page 8: "The regulon activities are highly consistent between integrated and individual cell atlases...", as more than 80% of the genes are the same.

We thank the reviewer for pointing out the ambiguity in text, relating to GRN inference between integrated and individual atlases.

Although the integrated atlas is an aggregation of individual atlases, yet it is quite distinct considering technical features between cell groups (# of cells sampled, # of genes detected, sequencing depth, dropout rates etc.,; Fig. S1, S6 and S7).

Indeed, we consider 11,425 overlapping genes (genes detected in at least 10% of total cells) for GRN framework (using SCENIC) that includes GRNBoost (identifying regulons), RCisTarget (Database crossmatching of regulon-target pairs), and AUCell (scoring regulon activity across individual cells). While the regulon inference (GRNBoost) are identical for all atlases, the regulon composition (RCisTarget) and scoring (AUCell) differs between integrated and individual atlases.

In the revised ms, we have updated the text and especially figures (Fig. S1, S6 and S7) to highlight differences between integrated and individual atlases.

Gene regulatory inference for integrated and each atlas (using SCENIC)

1. GRNBoost/GENIE3: Finds co-expressed and correlated TF-gene pairs
2. RCisTargets: Prunes away co-expressed TF-gene pairs using annotated motif database
3. AUCell: Scores regulons (TF-gene pairs) within each single-cell

Integrated	TM-10x	TM-SS2	MCA
Tissues: 50	Tissues: 12	Tissues: 18	Tissues: 54
Single-cells (SC): 244,642 Pseudobulk (PB): 4,550	Single-cells (SC): 54,865 Pseudobulk (PB): 1,061	Single-cells (SC): 38,888 Pseudobulk (PB): 703	Single-cells (SC): 150,889 Pseudobulk (PB): 2,776
Cell types: 831 Features: 17,138	Cell types: 55 Features: 19,369	Cell types: 81 Features: 21,373	Cell types: 732 Features: 34,967
Regulons: 174	Regulons: 222	Regulons: 233	Regulons: 233

Fig. S1 (subpanel): Detailed workflow of atlas-scale GRN analysis.

Fig. S6 (subpanels): Technical differences and biases in mouse cell atlases. (A) Number of cells (first row), library size (i.e. sequencing depth; second row) and number of genes detected (third row) for individual and integrated atlas. Each point represents a single-cell and the metrics are stratified and coloured based on 7 cell groups and 55 cell types with the median numbers reported next to box plots. (D) Pie charts showing proportion of single- (top) and pseudobulk (bottom) cells across the 7 cell groups in individual and integrated cell atlas. Note: proportions are conserved between single- and pseudobulk cells.

R1.2: The authors state in the introduction that there are multiple methods to infer gene regulatory networks. We missed the motivation why the authors chose SCENIC.

This is a valid point and we thank the reviewer for raising this. In the revised ms, we have updated text (Page 7; also, below) to highlight the motivation of choosing SCENIC for this atla-scale analysis.

“To infer gene regulatory networks (GRNs), we applied SCENIC, a framework for network inference, reconstruction and clustering from scRNA-seq data [10]. The SCENIC framework is applied directly on single-cell expression matrix and combines three different approaches for regulon identification and activity. The three approaches consist of (i) ‘GRNBoost’ for identification of TFs and co-expressed genes from single-cell expression matrix, (ii) ‘RCisTarget’ for defining ‘regulons’ (i.e enriched and validated TFs with their direct downstream target genes containing annotated motif, and prunes co-expressed indirect targets), (iii) ‘AUCell’ for scoring regulon activity (RAS: regulon activity scores) in single-cells. Our motivation for using SCENIC for atlas scale regulon inference was three-fold. Firstly, SCENIC utilises GRNBoost, RCisTargets and AUCell to identify and score known, direct TF-target interactions while pruning away indirect and co-expressed TF-target links. The RCisTarget crossmatches regulons with known TF-target databases, as opposed to de-novo predictions. This allows inference of both TF-TF and TF-target relationships and scoring of TF-target relationships, unlike other approaches. Secondly, SCENIC does not require a single-cell trajectory/pseudotime, unlike other widely applied GRN methods [12], and therefore is well suited to cell atlas-scale analysis. Thirdly, SCENIC is widely used and GENIE3/GRNBoost are scored amongst the top reconstruction methods in recent benchmarking study [12]. Applying SCENIC, we identify 279 unique regulons, with >60% (174 regulons) shared across the three atlases (Fig 1B). The high degree of regulon overlap between the three atlases, in spite of technical differences highlights that single-cell regulatory state is predominantly governed by core set regulators and their activities within individual cells. A recent study also applied SCENIC, but only for MCA data using only the author assigned cell-type labels [21].”

We additionally performed head-to-head comparison between our regulon framework with a published GRN inference approach (bigScale2) for TM-10x atlas in Fig. S21.

A. Comparing GRN inference methods

B. Regulon composition similarity

Supplementary Figure 21: Comparing different GRN methods for atlas-scale analysis. (A) Overlap of regulons and target genes identified in this study (SCENIC: GRNBoost and RCisTarget) and repeating analysis with published GRN method (Ioconno et al 2015) for TM-10x mouse atlas. (B) Jaccard index highlighting the overlap between regulon composition inferred on TM-10x atlas by our framework and repeating analysis with published GRN method (Ioconno et al 2015).

R1.3: AUCell results lack directionality. There are other statistical approaches to analyse regulons that provide signed TF activities. Authors could compare to these, or at least elaborate on whether this could be relevant for their study.

We agree with the reviewer that while AUCell scores regulon activities, it does lack directionality of TF-TF crosstalk, and hence our integrated regulon network is undirected. However, our focus is to identify consensus and atlas-specific GRNs (and activity scores) across broad cell groups, in spite of sampling a variety of tissues and technical differences. We believe that statistical GRN methods that provide signed TF activities would be quite useful for studying cell types within a tissue or during development or differentiation. Additionally, we believe that a caveat of such approaches (benchmarked in PMID: 31907445) is a requirement of single-cell pseudotime or trajectory, unsuited for whole organism atlases.

Although, our network lacks TF-TF crosstalk directionality; the RCisTarget (within SCENIC) confers directionality of TF-target regulation (not TF-TF crosstalk). As in R1.1 response, we also perform head-to-head comparison with expression correlation-based bigScale2 GRN inference approach, which fails to capture several regulons identified by SCENIC.

R1.4: The authors state: "The cell-type separation was refined with pseudo-bulk cells and we robustly recover both general and specific regulons." - Given Figure 1 C and D, we are not entirely sure whether this statement is actually true. We would like to see a quantitative analysis to prove that using pseudobulk actually improved cell type separation.

We thank the reviewer for pointing this out. We have now included several qualitative and quantitative comparisons between single- and pseudobulk cells in the revised manuscript. These include (i) PCA projections and distance to cluster centroids (Fig. 1E), (ii) computing Adjusted Mutual Information (AMI) and completeness (Fig. 1F) (iii) Gini coefficient as a measure of regulon importance in classifying cell groups (Fig. 1G, S11A), (iv) Regulon composition similarity (Fig. 1E), (v) Silhouette score to compare clustering Fig. (S11B) and (vi) RAS correlation for all and individual cell groups (Fig. S11C).

Figure 1: (E) Principal component analysis (PCA) of matched single- and pseudobulk cells based on RAS across individual atlases and coloured by 7 cell groups (first two columns). For each of the 7 cell groups, we plotted cluster centroid (column 3) and connected single- (circles) and pseudobulk (asterisk). Box plots (column 4) represent Euclidean distance of individual single- and pseudobulk cells to respective cell group centroid. The distance is a measure of clustering i.e., larger distances in pseudobulk highlight well separated clusters compared to single-cells. (F) Different measures of cluster comparison (top: Adjusted mutual information, bottom: Completeness) between pseudobulk and single-cells across integrated and individual mouse atlases, considering 7 cell groups. (G) Distribution of Gini coefficients per regulon in pseudobulk and single-cells across integrated atlas, considering all 7 cell groups. The Gini coefficient is a measure of inequality i.e. whether individual regulons contribute to individual (smaller Gini) or multiple cell groups (higher Gini). The pseudobulk cells have higher Gini coefficients and tighter distributions compared to single-cells, that highlights pseudobulk regulons contribute and distinguish multiple cell groups. (H) Comparison of regulon composition between atlases (Pairwise Jaccard index) considering TM-10x as reference. Each dot represents a regulon and overlap of its target genes across 3 atlases. The shaded area represents 95% confidence interval from the linear regression line.

A. PB vs SC comparisons

B.

C. Integrated atlas (RAS correlation)

Supplementary Figure 11: RAS comparison between single- and pseudobulk cells. (A) Distribution of Gini coefficients per regulon in pseudobulk and single-cells across integrated atlas, stratified by individual cell groups. (B) Silhouette score comparing clustering between single- and pseudobulk cells across individual and integrated atlas. (C) RAS correlation between single and pseudobulk cells in all cells and stratified by individual cell groups. Error bars represent the standard deviation across single- and pseudobulk cells. The individual cell group correlation is significantly improved compared to global, which further validates our classification of 7 cell groups.

R1.5: The authors state: "The regulon activities are highly consistent between integrated and individual cell atlases, across single and pseudobulk cells (Supplementary Fig 8A-G, S9A-B)." - The authors support their statement with individual examples but also here we would like to see a comprehensively quantitative analysis.

We thank the reviewer for also pointing this out. In the revised ms, we highlight the technical feature differences between integrated and individual atlases (Fig. S6) and across reference cell groups (Fig. S7). This is also mentioned above in response to R1.1

Our ability to effectively distinguish cell groups using RAS on integrated and individual atlases is emphasized by (i) visual separation on UMAP (Fig. 1C & S9A, see also Fig. S1), (ii) comparing cell-to-cell correlations (Fig. 2A & S13A), (iii) comparing regulon-to-regulon correlations (Fig. 2B & S15A-C), (iv) comparing regulon-to-regulon correlations (Fig. 2C & S15D-F). We combine these in a single figure for the reviewers.

Figure: Comparison between integrated and individual atlases

Additionally, as mentioned in the responses to R1.4, we perform qualitative and quantitative comparisons between ‘single- and pseudobulk cells’ and between ‘integrated and individual atlases’, including computing Adjusted Mutual Information (AMI) and completeness (Fig. 1F). We also compute Silhouette score as a measure of clustering comparison between integrated and individual atlases (Fig. S11B)

Figure 1: (F) Different measures of cluster comparison (top: Adjusted mutual information, bottom: Completeness) between pseudobulk and single-cells across integrated and individual mouse atlases, considering 7 cell groups.

Figure S11: (B) Silhouette score comparing clustering between single- and pseudobulk cells across individual and integrated atlas.

R1.6: For the sake of transparency and reproducibility, the authors should make their analysis code publicly available. (e.g. via GitHub). They mention a GitHub repository in the paper, but no link is provided?

We thank the reviewer and apologise for not enabling the Github link with jupyter notebooks (incl. revision analysis), which can be found here: <https://github.com/Natarajanlab/Single-cell-regulatory-network>.

Minor comments

R1.M1: In addition to the review of GRN reconstruction by Fiers et al., also a recently published benchmark of those methods could be referred to - Pratapa et al., Nature Methods (<https://doi.org/10.1038/s41592-019-0690-6>)

We have cited the GRN benchmarking paper in revised ms.

R1.M2: Typo in Supplementary Figure 5 A and B: alllast -> at least

We have corrected the typo in text.

R1.M3: Related to Figure 1C and others: Even though the term RAS (regulon activity score) is defined in the main text and method section, it would be helpful to define this term also in the legend/caption.

We have now expanded the term RAS in figure legends

R1.M4: "The individual regulons, their compositions and activity scores are detailed in Supplementary Table 1" page 7 - This is a wrong reference, as Supplementary Table 1 contains the results from scMAP. Also, the activity scores are not reported but the module number/ID.

We thank the reviewer for pointing this out. The supplementary table 3 (SuppTable3.xlsx) contains the mean regulon activity scores from pseudobulk cells across integrated atlas.

R1.M5: Why is the pathway analysis (Supplementary Figure 11 B) performed on regulons and not on all genes within regulons as in the GO analysis?

We thank the reviewer for pointing this out. As the smaller modules only contain a handful of regulons, the GO analysis considering regulone alone (without their composition) did not produce any significant terms. Therefore, we considered both regulons and their composition for GO analysis, as reported in Fig. S14.

For pathway analysis, the regulons alone (without their composition) produced significant and enriched pathways. Considering regulons and direct targets, the pathways (reported in Fig. S14) are still enriched, but we observe several highly specialized pathways; which likely do not operate across multiple cell groups. Hence we avoid considering the direct targets.

We have updated the figure text for Fig. S14.

R1.M6 No logical order supplementary Figures. Supplementary Figure 3 B is referenced first (page 5)

We thank the reviewer for pointing this out and have updated the main text.

Reviewer #3 (Comments to the Authors (Required)):

In this paper, the authors use existing cell atlases in mouse to build regulatory networks. They do this by first applying scMap to identify similar cell populations in each atlas and then apply SCENIC to do network inference. Network inference is done on the entire merged dataset, on a downsampled version and also on each cell atlas. Results are compared based on the consistency of recovery of regulons, defined by transcription factors enriched in a set of co-regulated/co-expressed genes. Although the inference of regulatory networks and modules from these published compendia is interesting to the community, and the finding that cell types could be discriminated by the regulon community is insightful, the presented approach and analysis does not seem sound and there are several points that I think need additional explanation or analysis.

We thank the reviewer for his/her comments, acknowledging the power of regulatory inference in distinguishing individual cells across mouse atlases, as well as the usefulness of our framework to the community.

We would like to re-emphasize our motivation to extract and validate the regulatory crosstalk through integrating analysis of atlas-scale datasets, especially given the rapid development of major cell, tissue and organism level atlases.

R3.1: Definition of cell types. The authors don't really provide any detail about how they determined the generalized vocabulary of 55 cell types and species. They simply cite Supp Fig 3A and it is unclear how to understand the process from this figure.

We thank the reviewer for pointing this out and apologise for lack of clarity. The generalised vocabulary of 7 cell groups is manually devised considering the reference, but also the 831 unique

author-annotated cell-types. We chose TM-10x as a reference atlas for scMap projections, and further linked its 55 cell types to 7 cell groups. Subsequently for each atlas, we use scMap to project the author-annotated cell types to our reference cell groups.

In the revised ms, we have updated text (Page 5-6; also below) and figures; especially Fig. S1 to provide a detailed overview of analysis.

“We aimed to integrate the atlases to identify cell-type specific regulons and build a consensus regulon atlas (Fig 1A; Detailed workflow in Fig. S1). As each atlas samples different mouse tissues and scRNA-seq technologies (full length vs 3’end) to identify hundreds of varied cell types across cellular resolutions (discussed below), a fundamental challenge is to effectively link the original author’s cell-type annotation across cell atlases. We address the challenge of integrating cell-type classification by combining two complementary approaches. Firstly, we manually devised a generalised vocabulary consisting of broadly defined ‘7 cell groups’ for an standardise annotation between cell atlases (3 datasets). Secondly, we utilise scMAP, an unsupervised scRNA-seq cell projection method [23], to link the original author’s cell-type annotation across cell atlases (Supplementary methods). By utilizing Tabula Muris 10x (TM-10x) Chromium annotations as a reference and by combining both approaches, our generalised vocabulary contains ‘7 cell groups’ consisting of ‘55 reference cell-types’. The 7 cell groups include Immune (22 subgroups), Specialised (12 subgroups), Epithelial (7 subgroups), Stem (4 subgroups), Endothelial (4 subgroups), Basal (3 subgroups) and Blood (3 subgroups) (Fig. S2A). Subsequently, we applied our two-step approach to individual atlases i.e. TM-10x (Fig. S2B), Tabula Muris Smart-seq2 (TM-SS2; Fig. S3A), MCA (Fig. S3A-B) and as well as to all three atlases integrated together (Fig. S4A). Our approach allows us to build and link an integrated mouse atlas consisting of 831-author assigned unique cell-type labels from 50 tissues to a consensus of 55 reference cell-types and 7 cell groups (Fig. S4A, methods and Table S1).”

Supplementary Figure 1

A. Detailed analysis workflow

Supplementary Figure 1: Detailed workflow of atlas-scale GRN analysis.

To effectively integrate single-cell annotations across three atlases, we first manually devised 7 reference cell groups, chose TM-10x as a common reference and assigned the 55 cell types to 7 cell groups respectively. Next, we used scMAP to link each atlas (TM-SS2 and MCA) to the reference and built an integrated mouse atlas with common vocabulary for all single-cells. Using a stringent feature selection cutoff, we performed gene regulatory network inference using SCENIC. This briefly includes TF and TF-target identification from single-cell expression matrices (GRNBoost), cross-validation of TF and its direct targets (i.e. Regulons) using annotated motif databases and pruning away indirect, co-expressed genes (RCisTargets) and lastly scoring the regulon activity (RAS: regulon activity score) within each single-cell. We applied the framework to integrated and individual atlases to (i) classify individual and pseudobulk cells based global regulon activity (UMAP), (ii) classify cells based on shared and distinct regulon activity (cell-

to-cell correlation), (iii) identify consensus and cell group specific regulons (regulon-to-regulon correlation) and (iv) build an atlas-scale regulon activity network.

R3.2: The authors claim that they are able to robustly map author specified cell types, but I am finding the figures hard to read; there is too much cross-edges between the cell types across the different datasets. It might be good to quantify this.

We thank the reviewer for the suggestion. As highlighted in the main text, we used scMap (with stringent 0.7 cutoff) for projecting atlases onto each other, considering TM-10x (55 cell types) as reference. We provide a contingency table (Table S1) containing mapping the TM-10x (55 cell types) to TM-SS2 (81 cell types), and MCA (732 cell types) respectively.

Additionally, we perform several qualitative and quantitative comparisons for individual atlases considering 7 reference cell groups. These include (i) PCA projections and distance to cluster centroids (Fig. 1E), (ii) computing Adjusted Mutual Information (AMI) and completeness (Fig. 1F) and (iii) Regulon composition similarity (Fig. 1E, S11B).

Figure 1: (E) Principal component analysis (PCA) of matched single- and pseudobulk cells based on RAS across individual atlases and coloured by 7 cell groups (first two columns). For each of the 7 cell groups, we plotted cluster centroid (column 3) and connected single- (circles) and pseudobulk (asterisk). Box plots (column 4) represent Euclidean distance of individual single- and pseudobulk cells to respective cell group centroid. The distance is a measure of clustering i.e., larger distances in pseudobulk highlight well separated clusters compared to single-cells. (F) Different measures of cluster comparison (top: Adjusted mutual information, bottom: Completeness) between pseudobulk and single-cells across integrated and individual mouse atlases, considering 7 cell groups. (G) Distribution of Gini coefficients per regulon in pseudobulk and single-cells across integrated atlas, considering all 7 cell groups. The Gini coefficient is a measure of inequality i.e. whether individual regulons contribute to individual (smaller Gini) or multiple cell groups (higher Gini). The pseudobulk cells have higher Gini coefficients and tighter distributions compared to single-cells, that highlights pseudobulk regulons contribute and distinguish multiple cell groups. (H) Comparison of regulon composition between atlases (Pairwise Jaccard index) considering TM-10x as reference. Each dot represents a regulon and overlap of its target genes across 3 atlases. The shaded area represents 95% confidence interval from the linear regression line.

R3.3: Although the authors say they are able to define matching cell types, I feel that using one or two additional approaches to correct for batches could be beneficial. E.g. conos, scanorama, liger, seurat are pretty standard and new approaches that people have applied and compared and should be used to verify their results.

We thank the reviewer for raising this point and were intrigued by the suggestion. The SCENIC authors highlight the robustness of GRN inference and scoring to overcome batch effects. To address the point for atlas scale data, we performed batch-correction using two widely used published methods (BBKNN and MNN-correct), considered a matched tissue (Spleen) from TM-10x and TM-SS2 data. Repeating regulon scoring, we observe highly similar and correlated regulon activity scores (RAS) and composition similarity between non-corrected and batch-corrected data.

In the revised ms, we have updated text (Page 10; also below) and figure (Fig. S12A-B) to report comparison of batch analysis.

“Given the different technical differences between individual atlases (Dropouts, tissues profiled, scRNA-seq protocol, sequencing depth etc.), we also assessed whether batch effects confound RAS across mouse atlases. Although SCENIC analysis has been shown to be unaffected by batch and technical effects [10], we performed batch correction on a common tissue (Spleen) profiled by both TM-10x and TM-SS2 atlases. We apply two methods ‘Batch-balanced KNN’ (BBKNN) and ‘Mutual nearest neighbors correction’ (MNN) [31,32] and visualise individual cells on t-distributed stochastic neighbor embedding (tSNE). The batch correction had minimal impact on resolving and overlapping similar cell types between the two atlases, compared to uncorrected data (Fig. S12A). Notably, the corrected batch effects were unique to each method on tSNE space. Performing SCENIC on uncorrected and two batch-corrected datasets, we find that individual regulon activities (RAS similarity) and regulon compositions (Jaccard coefficient) are highly correlated, indicating that batch effects have little effect on regulon activity (Fig. S12C-D). In summary, the pseudobulk approach accounts for technical and batch effects, robustly reports on regulon activities and leads to better classification of cell groups across individual and integrated atlas.”

A. Batch effect correction (expression space)

Supplementary Figure 12: Impact of batch effect correction on regulon inference. (A) UMAP embedding of pseudobulk cells from TM-10x and TM-SS2 atlases, considering either uncorrected or two batch corrected expression space (BBKNN and MNN-correct). The pseudobulk cells are coloured by cell atlas (top) and cell groups (bottom). Note: Both batch correction methods slightly improve the overlap of reference cell types compared to uncorrected UMAP. However, the clusters from BBKNN and MNN-correct don't overlap with each other and introduce additional discrepancies. **(B)** Pairwise correlation of individual regulons (based on RAS) from both batch correction methods compared to uncorrected data. Each dot represents a regulon identified in all 3 SCENIC runs (uncorrected, BBKNN and MNN-correct). The shaded area represents the 95% confidence interval from the linear regression line. **(C)** Regulon composition similarity computed from pairwise Jaccard index between batch corrected (BBKNN and MNN-correct) to uncorrected data. The shaded area represents the 95% confidence interval from the linear regression line.

R3.4: The definition of a regulon and its comparison across datasets needs to be more precise. Is a module, a set of co-expressed genes or co-expressed and co-regulated, or co-regulated? They define a regulon as "(modules of enriched TFs and direct regulators), which would suggest that a regulon is defined by group of regulators, but the downstream analysis only uses one TF at a time.

We thank the reviewer for the suggestion and apologise for the lack of clarity.

A regulon is simply a collection, which contains a single transcription factor (TF) and all its transcriptional target genes and includes both direct and indirect targets. Almost all regulon inference approaches first compute correlation for given dataset (bulk or single RNA-seq, ATAC etc.), and secondly identify/infer regulons with both correlated and co-regulated targets (direct and indirect). In the SCENIC framework, we first identify/infer regulons using GRNBoost. Critically, we next use RCisTarget to crossmatch identified regulons with annotated database of TFs and direct targets, to prune away indirect targets. Lastly, the regulons (TFs and direct targets) are scored in individual cells and provides regulon activity scores (RAS).

In revised text, we have clarified the definition of regulon (Introduction Page 4, and below).

A GRN is specific combination of transcription factors (TFs) and co-factors that interact with cis-regulatory genomic regions to mediate a specialised transcriptional programme within individual cells [9,10]. Briefly, a regulon is collection of a single transcription factor (TF) and all its transcriptional target genes. The GRNs define and govern individual cell-type definition, transcriptional states, spatial patterning and responses to signalling, cell fate cues [11].

Additionally, further clarified the SCENIC steps (Results Page 7, and below).

“To infer gene regulatory networks (GRNs), we applied SCENIC, a framework for network inference, reconstruction and clustering from scRNA-seq data [10]. The SCENIC framework is applied directly on raw uncorrected single-cell expression matrix and combines three different approaches for regulon identification and activity. The three approaches consist of (i) ‘GRNBoost’ for identification of TFs and co-expressed genes from single-cell expression matrix, (ii) ‘RCisTarget’ for defining ‘regulons’ (i.e enriched and validated TFs with their direct downstream target genes containing annotated motif, and prunes co-expressed indirect targets), (iii) ‘AUCell’ for scoring regulon activity (RAS: regulon activity scores) in single-cells. Our motivation for using SCENIC for atlas scale regulon inference was four-fold. Firstly, SCENIC utilises GRNBoost, RCisTargets and AUCell to identify and score known, direct TF-target interactions while pruning away indirect and co-expressed TF-target links. The RCisTarget crossmatches regulons with known TF-target databases, as opposed to de-novo predictions and allows inference, scoring of both TF-TF and TF-target relationships, unlike other approaches. Secondly, SCENIC does not require a single-cell trajectory/pseudotime, unlike other widely applied GRN methods [12], and therefore is well suited to cell atlas-scale analysis. Thirdly, SCENIC is widely used and GENIE3/GRNBoost are scored amongst the top reconstruction methods in recent benchmarking study [12].”

R3.5: Furthermore, the authors use a measure "Correlation Specificity Index" to define similarity between modules to examine the similarity of modules, but this is not well-defined. They mention Pearson correlation between regulons, but the regulon is a collection of genes and TFs/regulators. Hence using some mathematical notation could be beneficial here.

This is a good point and we thank the reviewer. Firstly, we acknowledge the typo as CSI refers to the Connection Specificity Index (as opposed to Correlation). The CSI is a graph metric (described

in PMID: 24296474) that ranks the regulon significance based on similarity and specificity of interaction partners. The CSI is calculated as:

The CSI for two nodes A and B is calculated by:

$$CSI_{AB} = 1 - \frac{\#nodes\ connected\ to\ A\ or\ B\ with\ PCC \geq PCC_{AB} - 0.05}{n_y}$$

Where the Pearson correlation coefficient (PCC) is the interactional correlation between A and B.

We have updated text (Page 11; also below) and stated the CSI notation in both figure legends (Fig. S15) and supplementary methods.

R3.6: The authors they use a "variety of different feature sets" and cite Supp Fig 5A-D. But all this shows is a set of venn diagrams and it is not clear what the criteria is for using a gene set. I also did not understand what the authors are showing in supp 5E. They say "The regulon definition was highly similar with differential gene composition, owing to variable sequencing (Supplementary Fig 5E)" I am not sure what is meant by "Variable sequencing" and I am not convinced the regulons are similar. It seems only the number of genes per regulator is plotted, but it does not inform us about the composition of the target set.

We thank the reviewer for raising this valid point. The individual atlases have several technical differences with respect to each other. These include different number of cells profiled, number of tissues profiled, Full-length vs 3' scRNA-seq chemistry and platform, dropout rates, sequencing depth as well as many unknown factors.

In revised ms, we summarise and highlight these differences in text (Page 6; also below) and figures (Fig. S1, S6 and S7).

“The individual atlases have technical difference owing to different number of cells profiled (S6A top panel), sequencing depth (library size, S6A middle panel), number of tissues profiled (12 TM-10x, 18 TM-SS2, 38 MCA; S2B, S3A-B), scRNA-seq chemistry (Full-length vs 3'), scRNA-seq platform and number of genes detected (S6A bottom panel). The dropout distribution for individual atlases highlights the relationship between number of cells profiled, library size and genes detected (Fig. S6B). Specifically, MCA compared to Tabula Muris atlases has the highest number of profiled cells at sparse sequencing depth, lower gene detected and highest dropout rates across reference cell groups (S6A-B). Our 7 reference cell groups and high and proportional number of cells from both integrated (Fig. S6C) and individual atlas (Fig. S6D). For example, the Immune cell group consists of 20,133 individual cells classified across 22 reference cell types, while the blood cell group consists of 1559 cells classified into 3 reference cell types (Fig. S6C and S2A). We further present the different technical features for each reference cell type across integrated and individual atlas (Fig. S7A). Our two-step approach consisting of simplified cell group and subgroup classification allows us to mitigate technical and cell-type label discrepancies, integrate mouse cell atlases to investigate global and specific regulators across atlases.”

Fig. S6 (subpanels): Technical differences and biases in mouse cell atlases. (A) Number of cells (first row), library size (i.e. sequencing depth; second row) and number of genes detected (third row) for individual and integrated atlas. Each point represents a single-cell and the metrics are stratified and coloured based on 7 cell groups and 55 cell types with the median numbers reported next to box plots.

We also agree with the reviewer that Venn diagram comparison of multiple feature sets can be confusing, as we present regulon inference results from a single feature set (minimum 10% genes detected). To simplify, we have removed the comparison of different feature sets in the revised text.

R3.7 & R 3.8

R3.7: The claim that "groups have good separation based on regulon activity scores" needs to be better quantified. They are using the original cell groups and the cell type labels to color the cells in UMAP coordinates. This grouping could be better quantified by clustering and checking if the clusters do correspond to the cell types.

R3.8: I did not see the relevance of the downsampled data analysis and the authors don't do a systematic comparison of whether the results are actually the same or different. They say that cell types are more refined and they again find global and specific regulons, but this is very qualitative and more principled comparisons are needed.

We thank the reviewer for raising these valid suggestions and apologise for the lack of quantitative comparisons. Indeed the crowding of points on UMAP makes visual comparison very tricky. This concern is also shared by reviewer 1.

We would like to clarify that the pseudobulk cells are an aggregate of 50 individual cells, based on original authors tissue and cell-type labels. These are not downsampled data/cells, rather robust cells with decreased expression noise and improved RAS. The pseudobulk also improves the integrated regulon atlas, as rare/sparse cell populations have diminished representation.

To address the reviewers points (As in R1.4 response), we have now included several qualitative and quantitative comparisons between single- and pseudobulk cells in the revised manuscript. These include (i) PCA projections and distance to cluster centroids (Fig. 1E), (ii) computing Adjusted

Mutual Information (AMI) and completeness (Fig. 1F) (iii) Gini coefficient as a measure of regulon importance in classifying cell groups (Fig. 1G, S11A) and (iv) Regulon composition similarity (Fig. 1E, S11B).

Figure 1: (E) Principal component analysis (PCA) of matched single- and pseudobulk cells based on RAS across individual atlases and coloured by 7 cell groups (first two columns). For each of the 7 cell groups, we plotted cluster centroid (column 3) and connected single- (circles) and pseudobulk (asterisk). Box plots (column 4) represent Euclidean distance of individual single- and pseudobulk cells to respective cell group centroid. The distance is a measure of clustering i.e., larger distances in pseudobulk highlight well separated clusters compared to single-cells. (F) Different measures of cluster comparison (top: Adjusted mutual information, bottom: Completeness) between pseudobulk and single-cells across integrated and individual mouse atlases, considering 7 cell groups. (G) Distribution of Gini coefficients per regulon in pseudobulk and single-cells across integrated atlas, considering all 7 cell groups. The Gini coefficient is a measure of inequality i.e. whether individual regulons contribute to individual (smaller Gini) or multiple cell groups (higher Gini). The pseudobulk cells have higher Gini coefficients and tighter distributions compared to single-cells, that highlights pseudobulk regulons contribute and distinguish multiple cell groups. (H) Comparison of regulon composition between atlases (Pairwise Jaccard index) considering TM-10x as reference. Each dot represents a regulon and overlap of its target genes across 3 atlases. The shaded area represents 95% confidence interval from the linear regression line.

A. PB vs SC comparisons

B.

C. Integrated atlas (RAS correlation)

Supplementary Figure 11: RAS comparison between single- and pseudobulk cells. (A) Distribution of Gini coefficients per regulon in pseudobulk and single-cells across integrated atlas, stratified by individual cell groups. (B) Silhouette score comparing clustering between single- and pseudobulk cells across individual and integrated atlas. (C) RAS correlation between single and pseudobulk cells in all cells and stratified by individual cell groups. Error bars represent the standard deviation across single- and pseudobulk cells. The individual cell group correlation is significantly improved compared to global, which further validates our classification of 7 cell groups.

R3.9: Similarly, I found the analysis of the individual atlases not as insightful. It was hard for me to see from Supp Fig 8,9 how we can infer consistency since these are different umap plots and we can really compare these projections. The cells could again be clustered and cluster-cell type

association could be established and the regulons could be compared thereafter, or even, without clustering.

We believe that independent analysis of integrated and individual atlas uncovers distinct features of the regulatory landscape. As in response to R1.5, we combine and present the visual and qualitative features of integrated and individual atlases in a single figure for the reviewers. These include (i) UMAP embedding (Fig. 1C & S9A, see also Fig. S1), (ii) comparing cell-to-cell correlations (Fig. 2A & S13A), (iii) comparing regulon-to-regulon correlations (Fig. 2B & S15A-C), (iv) comparing regulon-to-regulon correlations (Fig. 2C & S15D-F).

Figure: Comparison between integrated and individual atlases

For more qualitative and quantitative comparison between ‘single- and pseudobulk cells’ and between ‘integrated and individual atlases’, we compute Adjusted Mutual Information (AMI) and completeness (Fig. 1F) to the ground truth (i.e 7 cell groups). Both metrics show good agreement between single- and pseudobulk cells across both integrated and individual atlases.

Figure 1: (F) Different measures of cluster comparison (top: Adjusted mutual information, bottom: Completeness) between pseudobulk and single-cells across integrated and individual mouse atlases, considering 7 cell groups.

Lastly, we compare and speculate on specific regulon crosstalk identified only across individual atlases. Across the integrated network, we can broadly observe the smaller modules (1&5) bridge the connection between larger modules (3&4). However the MCA network clearly distinguishes the interaction specifically mediated by module 1 and module 3.

Figure: Comparison of regulon module crosstalk between integrated and MCA.

R3.10 & R3.11

R3.10: The Irf8 mutant versus wild type analysis again seems disconnected and does not naturally follow from the cell-atlas regulon analysis. Irf8 was one of regulators, but there were several others that were discussed. Furthermore, the targets of Irf8 inferred in the cell-atlas were not actually validated. Rather, the new scRNA-seq dataset was used to redefine modules and Irf8 was found as a regulator here.

R3.11: In general, there is no computational or experimental validation of the regulator-target/network relationships. It is not clear how accurate the inferred networks are. Additional comparison to existing databases of TF-target relationships is needed to support the inferred GRNs.

We thank the reviewer for both the valid comments. Indeed we highlight several regulons (including *Irf8*), which have fundamental implications for individual cell types across atlas. Yet, the single-cell data with IRF8 knockout (*IRF8*^{-/-} or KO) across myeloid differentiation provides a biological validation of regulon importances and RAS for cellular states. Furthermore, comparing IRF8 KO to wildtype cells, we observe minimal reduction in scRNA-seq expression but highly diminished RAS; indicative of the cell state switch from monocytes to granulocytes.

Nonetheless, we have performed additional validation of regulon network including (i) TFBS motif comparison and correlation within each module, (ii) validating regulon network with experimentally annotated STRING protein-protein interactions, (iii) validating regulon interactions (CSI) with experimentally annotated STRING protein-protein interactions, and (iv) validating regulon network with essential genes from Online Gene essentiality database

A. Overrepresented motifs per module

B. Protein-protein interactions (SPRING)

Total network edges: 4032
Edges identified in SPRING: 2290 (57%)

E. Essential Genes (OGEE)

Total network nodes (regulons): 174
Network nodes in OGEE: 154 (109 essential, 45 non essential)
Network nodes not in OGEE: 20

C. GRN network links also predicted in STRING

D. Regulon activity (CSI) vs STRING confidence

Supplementary Figure 17: Validation of regulon network. (A) Top: Motif correlation between individual regulons within each module. The rows and columns indicate individual motif sequences of different lengths. Bottom: representative examples of TFs and their enriched motifs for each regulon module. (B) Annotated protein-protein interactions (PPI) from STRING overlaid on integrated regulon network. STRING contains all regulons (nodes), and only STRING validated interactions (black edges) are highlighted in the regulon network. Over 55% of regulon network edges are validated by STRING. (C) Distribution of the STRING validated interactions captured in

regulon network, plotted across 20 percentile combined score bins (x-axis). The number of regulon network links are listed above individual bins. The combined score is a measure of confidence of STRING PPI. (D) Correlation between regulon connection specificity index (CSI) and STRING confidence score. The error bars represent the 95% confidence interval. Red line indicates the CSI threshold used to construct regulon network. (E) Regulon network overlaid with experimentally validated and essential genes (OGEE essentiality status). The enlarged nodes represent essential genes, while diminished nodes are non-essential. The regulons absent in OGEE are greyed out in the network.

Minor:

R3.M1: Euclidian should be Euclidean.

We have corrected the typo in text.

August 10, 2020

RE: Life Science Alliance Manuscript #LSA-2020-00658R

Dr. Kedar Nath Natarajan
University of Southern Denmark
Biochemistry and Molecular Biology
Campusvej 55
Odense 5230
Denmark

Dear Dr. Natarajan,

Thank you for submitting your revised manuscript entitled "Predicting gene regulatory networks from cell atlases". We would be happy to publish your paper in Life Science Alliance pending final revisions necessary to meet our formatting guidelines.

- please address the remaining concerns of Reviewer 1 and provide a point-by-point response to his/her comments
- please provide both your main and supplementary figures as separate files
- please add 'Fig.' in front of callout to S6A (Page 6)
- please double-check your figure callouts--in the manuscript text, there is a callout for Figure S7 B,D but these panels are neither in the figure nor in the legend; there is a callout for S12C-D, but figure and legend does not have panel D (page 10)
- please add a figure callout for S12B and S22C
- please add the figure legends to the main manuscript text
- please remove Panel A for Fig. S1, Fig. S4, Fig. S7
- please provide more descriptive legends for Fig. S18, S19, and S20
- please enhance the resolution for figures S2, S3, S4, S5, S21 to ensure that the text is readable; each figure needs to fit on one page, so you might have to split figures into 2 figures
- for Fig. S3A: we suggest to make at least the text showing the individual tissues and cell groups larger
- for Fig. S3B and Fig. S4: Please ensure this figure is legible. It might be best to remove the text on "Authors cell types". The text describing tissues, reference cell types and cell groups should be enlarged or alternatively, you could work with color codes and a legend.
- for Fig. S21 A and B: these could be enlarged or maybe even placed on two different pages.

A. FINAL FILES:

B. MANUSCRIPT ORGANIZATION AND FORMATTING:

Sincerely,

Reilly Lorenz
Editorial Office Life Science Alliance
Meyerhofstr. 1
69117 Heidelberg, Germany
t +49 6221 8891 414
e contact@life-science-alliance.org
www.life-science-alliance.org

Reviewer #1 (Comments to the Authors (Required)):

Notes to the authors:

We thank the authors for their detailed answers to our raised points and the additional analyses. We feel that there are still a few open points not properly addressed, but this is due likely due to a misunderstanding - which might also be partly our fault due to not precisely asked questions. We would be happy to revise a short further round for the points below.

Major comments:

R1.3 We have the feeling that the authors mixed up the process of GRN reconstruction and the final inference of TF activities based on our question. The term "directionality" was intended to be related to the TF activity and not to the TF-target interactions. We apologize for not being more precise in our comment.

Both processes (GRN construction and TF activity inference) can be executed independently from each other. This means for example that the authors still could use Genie3 and RcisTarget from the SCENIC workflow for network reconstruction without relying on pseudotime or trajectories information. For the TF activity inference, AUCell might be replaced with any other statistical method that aims to analyze gene sets and is suited for single-cell data. VIPER (<https://www.bioconductor.org/packages/release/bioc/html/viper.html>) is one of these methods that returns for each TF a signed normalized enrichment score which is considered a proxy of TF activity. Our point in the revision is that we believe that this information might be superior to the undirected AUCell output (only values from 0-1).

R1.4 We thank the authors for the vast number of additional analyses aiming to support the initial hypothesis that cell types are better separated in reduced dimensionality space using pseudobulk approach instead of the original single-cell data.

However, we are confused with Figure 1E. We understand that the authors determined separately for single-cell and pseudobulk the centroids for each cell type cluster. Subsequently, the distance from the individual cells/meta-cells is computed to their respective cluster centroid. We believe that a large distance means that the cluster members are less tight arranged and thus spread around in space while a small distance corresponds to a relatively narrow cluster, which goes along with a better separation among the individual clusters/cell types.

However, the authors connect larger distances in the pseudobulk approach with a better separation of cell types. ("The distance is a measure of clustering i.e., larger distances in pseudobulk highlight well-separated clusters compared to single-cells"). We could not understand this conclusion.

Minor comments:

1) The motivation why the authors used SCENIC could be more concise. Even though we asked for this we believe that half a page is not needed, a couple of lines are enough. Especially, the detailed summary of SCENICs workflow could either go to supplement or be removed.

2) The authors missed answering to our 3rd minor comment.

"The GRN from SCENIC covers consistently across all atlases only a limited number of TFs/regulons (279). Could the authors elaborate on what could be the reason for this?"

3) There are extra parenthesis and the end of the 2nd paragraph of the introduction. Probably an artifact from the previous citation style.

4) Regarding Figure S11B. The scale of the silhouette plots should be the same for the single-cell and pseudobulk approach to facilitate the comparison.

5) We thank the authors for making their code publicly available. We would welcome it if the authors could provide some information in the README file to guide through the structure of the repository.

Reviewer #3 (Comments to the Authors (Required)):

The authors have done a good job of addressing my concerns. The only thing I would mention is that the computational validation of the regulatory network is happening at the regulon-regulon level, which one can think of as a TF-TF network, rather than a TF-gene network (which includes TF-TF edges). It would be good to make this clear in the paper that this is what the paper is ultimately finding. I also think that the co-presence of motifs is circular because SCENIC uses motifs to filter edges. What I was looking for is a validation of some of the well-known cell types with a key TF, with available CHIP-seq data, e.g. from ENCODE. But this is a "nice to have" point and the authors are not required to do this for the publication.

BLACK = reviewers comments

BLUE = our response

RED = revised text

Reviewer #1 (Comments to the Authors):

We thank the authors for their detailed answers to our raised points and the additional analyses. We feel that there are still a few open points not properly addressed, but this is due likely due to a misunderstanding - which might also be partly our fault due to not precisely asked questions. We would be happy to revise a short further round for the points below.

We thank the reviewer for his/her comments for the second review, which has taken a significant additional time. Also acknowledging that we have addressed the raised points and critically performed several additional analyses to strengthen the manuscript.

Major comments:

R1.3 We have the feeling that the authors mixed up the process of GRN reconstruction and the final inference of TF activities based on our question. The term "directionality" was intended to be related to the TF activity and not to the TF-target interactions. We apologize for not being more precise in our comment. Both processes (GRN construction and TF activity inference) can be executed independently from each other. This means for example that the authors still could use Genie3 and RcisTarget from the SCENIC workflow for network reconstruction without relying on pseudotime or trajectories information. For the TF activity inference, AUCell might be replaced with any other statistical method that aims to analyze gene sets and is suited for single-cell data. VIPER (<https://www.bioconductor.org/packages/release/bioc/html/viper.html>) is one of these methods that returns for each TF a signed normalized enrichment score which is considered a proxy of TF activity. Our point in the revision is that we believe that this information might be superior to the undirected AUCell output (only values from 0-1).

We thank the reviewer for his/her comments, and acknowledging the potential ambiguity. We have now included a comparison between AUCell and VIPER for B-cells from the TM-10x cell atlas.

A. Comparing GRN Scoring
(AUC vs NES)

B. Comparing GRN Scoring
(RSS vs NES)

Supplementary Figure 22: Comparing different GRN scoring methods

A. Correlation between AUCell and VIPER scoring of regulons across B-cells from TM-10x atlas. For each regulon, x-axis represents Area Under the Curve (AUC) computed by AUCell, while y-axis represents normalised enrichment score (NES) computed by VIPER.

B. Correlation between regulon specificity score (RSS) and NES (VIPER) for each regulon across B-cells from TM-10x atlas.
Each dot represents a regulon and the shaded area represents 95% confidence interval from the linear regression line.

R1.4 We thank the authors for the vast number of additional analyses aiming to support the initial hypothesis that cell types are better separated in reduced dimensionality space using pseudobulk approach instead of the original single-cell data.

However, we are confused with Figure 1E. We understand that the authors determined separately for single-cell and pseudobulk the centroids for each cell type cluster. Subsequently, the distance from the individual cells/meta-cells is computed to their respective cluster centroid. We believe that a large distance means that the cluster members are less tight arranged and thus spread around in space while a small distance corresponds to a relatively narrow cluster, which goes along with a better separation among the individual clusters/cell types.

However, the authors connect larger distances in the pseudobulk approach with a better separation of cell types. ("The distance is a measure of clustering i.e., larger distances in pseudobulk highlight well-separated clusters compared to single-cells"). We could not understand this conclusion.

We thank the reviewer for his/her comments.

We interpret the distance from cluster centroid (single- and/or pseudo-bulk cells) as a measure of cell group separation, as mentioned in the figure legend and as the reviewer highlighted.

Additionally, we believe that the distance alone does not fully account for the spread of points (single- and/or pseudo-bulk cells). We highlight this over tSNE plotting RAS for the TM-10x cell groups, where we can appreciate better separation of different cell groups (& reference cell types) including basal, stem and blood cell groups. We also provide the pseudobulk and single-cells RAS UMAPs in Fig. S9A-B.

We have removed the above statement from the figure legend to avoid confusion.

Minor comments:

1) The motivation why the authors used SCENIC could be more concise. Even though we asked for this we believe that half a page is not needed, a couple of lines are enough. Especially, the detailed summary of SCENICs workflow could either go to supplement or be removed.

We've updated the text accordingly.

2) The authors missed answering to our 3rd minor comment.

"The GRN from SCENIC covers consistently across all atlases only a limited number of

TFs/regulons (279). Could the authors elaborate on what could be the reason for this?"

We apologise for this.

We start with ~650 TFs, based on SCENIC and first filter TFs and genes not expressed in 10% of the cells (Fig. S1). The remaining ~450 TFs undergo second filtering step with GRNBoost (co-expression) and RCisTarget (TF motif). The resulting 279 regulons are used for scoring and analysis (Fig. S1)

3) There are extra parenthesis and the end of the 2nd paragraph of the introduction. Probably an artifact from the previous citation style.

We've corrected the text.

4) Regarding Figure S11B. The scale of the silhouette plots should be the same for the single-cell and pseudobulk approach to facilitate the comparison.

We've updated the figure.

5) We thank the authors for making their code publicly available. We would welcome it if the authors could provide some information in the README file to guide through the structure of the repository.

We've now added a README with the structure of the repository.

Reviewer #3 (Comments to the Authors (Required)):

The authors have done a good job of addressing my concerns. The only thing I would mention is that the computational validation of the regulatory network is happening at the regulon-regulon level, which one can think of as a TF-TF network, rather than a TF-gene network (which includes TF-TF edges). It would be good to make this clear in the paper that this is what the paper is ultimately finding. I also think that the co-presence of motifs is circular because SCENIC uses motifs to filter edges. What I was looking for is a validation of some of the well-known cell types with a key TF, with available ChIP-seq data, e.g. from ENCODE. But this is a "nice to have" point and the authors are not required to do this for the publication.

We thank the reviewer for his/her comments and appreciate inputs for improving our manuscript.

We are glad to have addressed all the concerns.

August 31, 2020

RE: Life Science Alliance Manuscript #LSA-2020-00658RR

Dr. Kedar Nath Natarajan
University of Southern Denmark
Biochemistry and Molecular Biology
Campusvej 55
Odense 5230
Denmark

Dear Dr. Natarajan,

Thank you for submitting your Research Article entitled "Predicting gene regulatory networks from cell atlases". It is a pleasure to let you know that your manuscript is now accepted for publication in Life Science Alliance. Congratulations on this interesting work.

DISTRIBUTION OF MATERIALS:

Again, congratulations on a very nice paper. I hope you found the review process to be constructive and are pleased with how the manuscript was handled editorially. We look forward to future exciting submissions from your lab.

Sincerely,

Reilly Lorenz
Editorial Office Life Science Alliance
Meyerhofstr. 1
69117 Heidelberg, Germany
t +49 6221 8891 414
e contact@life-science-alliance.org
www.life-science-alliance.org